# Past and future interannual variability of Arctic sea ice in coupled climate models

John R. Mioduszewski[1], Stephen Vavrus[1], Muyin Wang[2, 3], Marika Holland[4], and Laura Landrum[4]

[1]Nelson Institute Center for Climatic Research, University of Wisconsin—Madison, Madison, Wisconsin.

[2]Joint Institute for the Study of the Atmosphere and Oceans, University of Washington, Seattle, Washington.

[3]Pacific Marine Environmental Laboratory, National Oceanic and Atmospheric Administration, Seattle, Washington.

[4]National Center for Atmospheric Research, Boulder, Colorado.

*Corresponding author:* Steve Vavrus, sjvavrus@wisc.edu

## Abstract

The diminishing Arctic sea ice pack has been widely studied, but mostly focused on time-mean
changes in sea ice rather than on short-term variations that also have important physical and so-
cietal consequences. In this study we test the hypothesis that future interannual Arctic sea ice ar-
ea variability will increase by utilizing 40 independent simulations from the Community Earth
System Model's Large Ensemble (CESM-LE) for the 1920-2100 period, and augment this with
simulations from 12 models participating in the Coupled Model Intercomparison Project Phase 5
(CMIP5). Both CESM-LE and CMIP5 models project that ice area variability will indeed grow
substantially, but not monotonically in every month. There is also a strong seasonal dependence
in the magnitude and timing of future variability increases that is robust among CESM ensemble
members. The variability in every month is directly correlated with the average ice retreat rate
before there is an eventual disappearance in both terms as the ice pack becomes seasonal in
summer and autumn by late century. The peak in variability correlates best with the total area of
ice between 0.2 - 0.6 m monthly thickness, indicating that substantial future thinning of the ice
pack is required before variability maximizes. Within this range, the most favorable thickness for
high areal variability depends on the season, especially whether ice growth or ice retreat process-
es dominate. Thermodynamic melting (top, bottom, lateral) and growth (frazil, congelation) pro-
cesses are more important than dynamical mechanisms, namely ice export and ridging, in con-
trolling ice area variability.


## 1. Introduction

Arctic sea ice extent has declined by more than 40% since 1979 during summer (e.g. Stroeve et al. 2012; Serreze and Stroeve 2015; Comiso et al. 2017), primarily as a consequence of greenhouse gas forcing (Notz and Marotzke 2012) but also internal variability (Ding et al. 2017). While this trend is greatest in summer, substantial losses are observed throughout the year (Cavalieri and Parkinson 2012) resulting in an ice season duration that is up to 3 months shorter in some regions (Stammerjohn et al. 2012). Reduced ice area is accompanied by a greater fraction of younger ice (Nghiem et al. 2006; Maslanik et al. 2007a, 2011), which reduces the mean thickness of the basin ice pack (Kwok and Rothrock 2009; Kwok et al. 2009; Lang et al. 2017). As a result, the estimated negative trend in sea ice volume (-27.9% per decade) is about twice as large as the trend in sea ice area (-14.2% per decade; Overland and Wang 2013).

Output from many climate models suggests that the Arctic sea ice cover will not retreat in a steady manner, but will likely fluctuate more as it diminishes, punctuated by occasional Rapid Ice Loss Events (RILEs; Holland et al. 2006; Döscher and Koenigk 2013). The overall decline in ice cover is expected to continue (Collins et al. 2013), and the Arctic may become seasonally ice-free within a few decades, depending on emissions pathway (Stroeve et al. 2007; Wang and Overland 2009; 2012; Massonnet et al. 2012; Wang and Overland 2012; Overland and Wang 2013; Jahn et al. 2016; Notz and Stroeve 2016). However, internal variability confounds prediction of this timing (Stocker et al. 2013; Swart et al. 2015; Jahn et al. 2016; Labe et al. 2018), and even the definition of ice-free differs among Arctic stakeholders (Ridley et al. 2016). Nonetheless, navigation through the Arctic has already increased in frequency as a result of this decline (Melia 2016; Eguíluz et al. 2016), and even more trade routes associated with the increased ice-free season are expected by the end of the 21$^{st}$ century (Aksenov et al. 2015; Stephenson and Smith 2013).

As the Arctic sea ice pack thins and retreats, multi-year ice is being lost and there is consequently a larger proportion of seasonal, thin first-year ice (Kwok et al., 2010, Maykut 1978; Holland et al. 2006). Overall thinner ice may result in an ice pack that exhibits greater inter-annual variability (Maslanik et al. 2007b; Goosse et al. 2009; Notz 2009; Kay et al. 2011; Holland and Stroeve 2011; Döscher and Koenigk 2013), at least partially due to enhanced ice growth and melt (Maykut 1978; Holland et al. 2006; Bathiany et al. 2016a). Decreased ice thickness promotes amplification of a positive ice-albedo feedback, which can magnify sea ice anomalies (Perovich et al. 2007), and thin ice is more vulnerable to anomalous atmospheric forcing and oceanic transport due to the smaller amount of energy required to completely melt the ice (Maslanik et al. 1996, Zhao et al. 2018). For example, pulse-like increases in oceanic heat transport can trigger abrupt ice-loss events in sufficiently thin ice (Woodgate et al. 2012).

Changes in the interannual variability of sea ice ehave been studied only in a limited capacity, likely because they are only beginning to become visible in September in the present day. Both Goosse et al. (2009) and Swart et al. (2015; their Fig. S6) reported that maximum ice area variability during September occurs once the mean ice extent declines to 3-4 million km$^2$. This increased variability may occur due to increased prevalence of RILEs and periods of rapid recovery during this timeframe (Döscher and Koenigk 2013). The thickness distribution during these periods skews toward thinner ice, which is conducive to both rapid ice loss and rapid re-

covery processes (Tietsche et al. 2011; Döscher and Koenigk 2013). Holland et al. (2008) con-
sidered a critical ice thickness that can serve as a precursor to RILEs, but found it more likely
that intrinsic variability played the primary role in the particular RILEs that were studied.  More
recently, Massonnet et al. (2018) analyzed the projected variability of sea ice *volume* and its pro-
jected future change in the CMIP5 ensemble, which suggested a monotonic future decrease.  The
corresponding variability of sea ice area was investigated by Olonscheck and Notz (2017), but
their analysis was much coarser temporally and seasonally, in that it only compared changes be-
tween entire blocks of time (the historical 1850-2005 period vs. the future 2006-2100 interval)
and was further restricted to the summer and winter seasons.
Building on these previous studies, our paper has two novel aspects. First, we analyze the
transient interannual variability of sea ice area over the course of the year from the early 20$^{th}$
century through the entire 21st century and find very different behavior across the four seasons.
These monthly differences are societally important, because marine access to the Arctic will like-
ly expand beyond late summer as the ice pack shrinks. Second, we detail how interannual sea ice
area variability changes as the ice pack retreats, and we link enhanced future variability to opti-
mal ice thicknesses and to the various thermodynamic and dynamic processes that control ice
area variability. We analyze a large 40-member ensemble from a single GCM, which allows us
to isolate internal variability, which is otherwise muddled with inter-model variability in multi-
model comparisons. This allows us to test the hypothesis that inter-annual Arctic sea ice cover
variability will increase throughout the year in the future as the ice pack diminishes.

## 2. Data and Methods

Ice thickness, concentration, and area were obtained from simulations of the Community
Earth System Model Large Ensemble Project (CESM-LE). Ice concentration refers to the per-
centage of a given grid cell that is covered by ice, while ice area in this study refers specifically
to this percent coverage multiplied by the area of the grid cell yielding a total Arctic ice-covered
area. The CESM-LE was designed to enable an assessment of projected change in the climate
system while incorporating a wide range of internal climate variability (Kay et al. 2015). It con-
sists of 40 ensemble members simulating the period 1920-2100 under historical and projected
(RCP8.5 emissions scenario only) external forcing. The ensemble members are produced by in-
troducing a small, random round-off level difference in the initial air temperature field for each
member. This then generates a consequent ensemble spread that is purely due to simulated inter-
nal climate variability. A full description of the CESM-LE is given in Kay et al. (2015), and sim-
ilar ensembles using the weaker RCP4.5 and RCP2.6 scenarios can be found in Sanderson et al.
133  (2017, 2018).

Another data set used in the current study is the model simulations from the Coupled
Model Intercomparison Project Phase 5 (CMIP5). Although more than 40 models submitted their
simulation results to the Program for Climate Model Diagnosis and Intercomparison (PCMDI),
only 12 of them simulated the Arctic sea ice extent both of the monthly means (each individual
month) and the magnitude of the seasonal cycle (March minus September sea-ice extent) within
20-percent error when compared with observations (Wang and Overland, 2012, Wang and Over-
land 2015). Therefore, we used only these 12 models identified by Wang and Overland (2015) in
this study: ACCESS1.0, ACCESS1.3, CCSM4, CESM1(CAM5.1), EC-EARTH, HadGEM2-
AO, HadGEM2-CC, HadGEM2-ES, MIROC-ESM, MIROC-ESM-CHEM, MPI-ESM-LR, and
MPI-ESM-MR. Among the 12 models, half of them use the same sea ice model as CESM-LE
(CICE, Hunke and Lipscomb 2010) or a variation of it. If a GCM provided multiple ensemble
members, we only kept up to 5 realizations, so that the total ensemble numbers is close to that
used in CESM-LE. There are a total of 33 ensemble members from these 12 models in the
RCP8.5 emissions scenario. Sea ice area, rather than ice extent, is computed from these 12
CMIP5 models to be consistent with CESM-LE results.
One of our primary analysis datasets is the time series of monthly ice variables. The en-
semble mean of all variables is taken after the statistics are calculated for each ensemble mem-
ber. 1-year differences in ice area are calculated for each month separately to remove the con-
founding effect of amplified variability resulting from a downward trend. Finally, a 10-year run-
ning standard deviation is applied to the time series of 1-year differences in monthly ice area,
centered on a given year. Ten years was chosen to quantify variability over decadal-scale inter-
vals and to provide an adequate number of years for a standard deviation calculation. The timing
and magnitude of variability is generally insensitive to the standard deviation window, however,
and whether the 1-year difference in ice area or its raw time series is used.
**3. Results**
**3.1     Sea ice area and its variability**
Sea ice area in the CESM-LE is projected to decline in all months in the 21$^{st}$ century,
proceeding in three phases: a fairly stable regime of extensive coverage in the 20$^{th}$ century, then
a decline, followed by virtually no ice remaining in summer and autumn months (Fig. 1). Sea ice
area variability follows an analogous three-phase progression in months spanning mid-summer
to early winter (Fig. 2). For example, in September this includes a period of modest variability
during the 20$^{th}$ century, then a distinct variability peak in the late 2020s and 2030s that coincides
with the maximum rate of ice retreat, and finally negligible variability in the late 21$^{st}$ century as
the Arctic reaches near ice-free conditions (Fig. 2). The first two phases of this progression in
variability occur for months in late winter to early summer (January-June), and suppressed varia-
bility would likely emerge beyond the end of the century, assuming that ice cover in these
months would continue to retreat. The maximum rate of ice retreat (negative values of the de-
rivative) occurs at a different time in the 21$^{st}$ century in each month, occurring presently in Sep-
tember but not until the end of the century in spring.
The same relationship between ice area and its variability is maintained across CMIP5
models, though with more noise resulting from the aggregation of many different models rather
than ensemble members from a single model (Fig. 3). This is most notable in the sea ice area (1-
year difference) time series (Fig. 3, blue), indicating that there is considerable spread in when
and how the downward trend proceeds each month, as found in Massonnet et al. (2012), but
good agreement that variability increases in this timeframe.

The analysis of ice area variability in Fig. 2 and Fig. 3 follows that of Goosse et al.
(2009) and Swart et al. (2015), but we extend their findings for September to all months and con-
firm that the variability in ice area is maximized as its total basin area decline is well underway
in both CESM-LE ensembles and across CMIP5 models. A directs relationship between the rate
of sea ice retreat and the magnitude of variability is present across all months in CESM-LE and
CMIP5: the standard deviation is highest when ice declines the fastest (Figs. 1 and 2). Further-
more, the magnitude and timing of peak ice area variability in both sets of experiments differs
greatly by season. The peak in magnitude in CESM-LE is most pronounced from November-
January when the running standard deviation of ice area exceeds $1 \times 10^6$ km$^2$, while the lowest
magnitudes occur in April and May, when the downward trend in ice area does not peak prior to
2100 (Fig. 2). Near the end of the 21$^{st}$ century, the running standard deviation also shows an in-
crease in the CMIP5 ensembles from December to June (Fig. 3), very similar behavior to that
displayed by CESM-LE. However, the magnitude of the increase in the running standard devia-
tion in the CMIP5 ensemble mean is smaller than that in CESM-LE. This is not surprising, as the
timing of ice retreat varies among models, so averaging them will smooth out the possible sig-
nals. The CMIP5 models therefore provide additional evidence that increased variability is asso-
ciated with decreasing sea ice coverage.
## 3.2 Relationship between ice area variability and thickness
Because increasing future concentrations of thin ice are likely a primary factor in in-
creased ice area variability, we next consider the relationship between ice thickness and ice area
variability in CESM-LE. This is done by correlating the standard deviation of basin-wide ice ar-
ea (Fig. 2) with the total area of grid cells with mean ice thickness within a given range for an
aggregation of all years and ensemble members, binned at 0.05 m intervals (Fig. 4). 20th century
data are omitted because both variables are largely stationary for this period. There is a large dif-
ference in the maximum correlation coefficient across seasons, but in most months it peaks be-
tween r = 0.6 and r = 0.8. This peak is associated with the thinnest ice of 0.1 m to 0.2 m from
October to January. There is a broad peak in the correlation coefficient between 0.25 m and 0.40
m in August and September, while July peaks near 0.45 m thickness but with a weaker maximum
correlation coefficient (r = 0.6). In June, r = 0.6 for most ice thicknesses below 0.8 m, and there
is only a weak correlation between these variables in April and May.
The analysis in Fig. 4 allows us to identify a common range of ice thicknesses when ice
area variability generally peaks regardless of the month, which we approximate as 0.2 m to 0.6
m. We next track the temporal evolution of this thin ice throughout the basin by calculating the
total area of ice that falls within that range. The time-transgressive nature of when the peak in
thin ice cover occurs (earliest in September, latest in winter-spring) is consistent with the corre-
sponding timing of the peak future sea ice area variability, suggesting that the emergence of a
sufficiently thin and contracted ice pack is a primary factor for enhanced ice cover variability
(Fig. 5). Both curves match each other in shape, with a steady state early, increasing to a peak
and dropping to zero as the Arctic becomes ice-free. The exception is in the spring and early
summer when neither increases until the end of the 21st century, when ice begins to decline more
rapidly. The two curves are largely in phase as well, with one preceding the other by no more
than 10-20 years in July, August, and November–January. The phase difference is due to the
chosen range of ice thicknesses, since the best relationship varies by month (Fig. 4). The two
curves are in phase from August-October (Fig. 5) when the 0.2 m to 0.6 m range approximates
the best relationship between thickness and variability (Fig. 4). However, ice area variability
maximizes after the peak in 0.2 m – 0.6 m thickness area in November–January, because varia-
bility is more highly correlated with ice slightly thinner than 0.2 m in these months (Fig. 4; Fig.
238  5).
There are also notable seasonal differences in the spatial pattern of variability during the
decade when variability in ice concentration peaks in CESM-LE (Fig. 6). The largest fluctuations
occur in a horseshoe-shaped pattern across the Arctic Ocean in autumn, but they are restricted to
the boundaries of the Atlantic and Pacific Oceans in late winter and spring. The result is a larger
area of high variability in the second half of the year and into January. The mean 0.2 m (dotted)
and 0.6 m (solid) ice thickness contours are overlaid for reference (Fig. 6). The contours corre-
spond closely to the boundary of maximum variability in ice coverage in most months, which is
consistent with results from Fig. 4 and Fig. 5. This demonstrates the first-order relationship be-
tween thin ice and the variability of inter-annual ice coverage within a given region.
**3.3 Ice concentration tendency**
The strong relationship between thin ice coverage and high concentration variability oc-
curs primarily due to the differing underlying mechanisms controlling ice concentration variabil-
ity at a given time, namely whether ice is expanding or retreating. To illustrate this, we chose
two months representative of these processes, September and December, to conduct an in-depth
analysis of the physical mechanisms involved in the time difference in the two curves in Fig. 5.
September is the end of the melt season, and therefore the ice concentration over the entire basin
in this month reflects the cumulative impact of melt processes throughout the summer. By con-
trast, December is a time of ice growth, particularly in the future, and thus the ice concentration
in this month is largely regulated by cumulative growth processes during the autumn. Using
available model output, we calculate the ice concentration tendency (% day$^{-1}$) from thermody-
namics and dynamics in the regions where the decadal standard deviation of ice concentration
exceeds 30% within the grid cell (Fig. S1) to evaluate the mean ice budget. These regions of
maximum variability in September and December closely match those in Fig. 6, though the mag-
nitude is smaller in Fig. 6 due to the standard deviation being a decadal mean. The daily change
in ice concentration is a function of dynamic contributions (ice import/export and ridging), ther-
modynamic melt processes (the sum of top, basal, and lateral), and thermodynamic growth (frazil
and congelation). Because antecedent conditions of the icepack can be an important factor for
determining ice concentration in the month of interest, we sum these terms over the preceding
months (July-September or October-December) and report the net 3-month change in ice concen-
tration resulting from each component.
The most interannually variable ice cover during September occurs primarily in the 2020s
and is centered across the central Arctic (Fig. S1a), though this region displays net ice expansion
in July-September in the 20th century (Fig. 7a) due to rapid ice growth in September. Thermo-
dynamic processes dominate over dynamics and are of opposing sign during the 20th century,
and thermodynamic processes add an average of 20% to the ice concentration of each grid cell in
the region by the end of September, compared with a loss of only 10% from dynamical processes
(Fig. 7a). Ice growth diminishes and melt processes accelerate in the early-mid 21st century
when the melt processes reduce ice concentration by more than 75% and the dynamic processes
essentially disappear with less ice to export (Fig. 7a). After 2060, September ice-free conditions
occur, and the thermodynamic term becomes less negative due to reduced areal coverage of ice
in June and hence less ice area to melt over the summer (Fig. 7a).
Because thermodynamic processes dominate in controlling ice concentration in the fu-
ture, they should also be the first-order forcing explaining future ice concentration variability,
particularly given that the magnitude of the dynamic contribution approaches zero by the 2020s
when ice cover is rapidly diminishing. As shown in Figure 7b, the peak interannual variability in
the thermodynamic term (red curve) is indeed several times larger than peak variability of the
dynamic term (blue curve), and the variability in the thermodynamic term maximizes during the
late 2020s in phase with the variability of the ice concentration (green curve) when the thermo-
dynamic term is declining most rapidly in Figure 7a. The variability likely also reflects the influ-
ence of the surface albedo feedback in amplifying summer ice area variations. There is a second-
ary rise in the variability of the thermodynamic term after 2060 (Figure 7b), coinciding with its
rapid rise toward zero in Figure 7a, but ice coverage by this point is confined to a diminishing
area.
From the 20[th] century well into the 21[st] century, ice growth occurs in the October-
December period in a similar region of maximum interannual variability as September, except
slightly equatorward (Fig. S1b). Ice export plays a relatively larger role in the regions of interest
in December than in September (Fig. 7c). However, the thermodynamic tendency is still the
dominant term controlling ice concentration within this region of maximum interannual variabil-
ity, and this term increases in the early-mid 21[st] century to a total of nearly 120%, some of which
is offset by ice export that contributes to a 40% decrease in mean ice concentration in the 20[th]
and early 21[st] centuries (Fig. 7c). The increased net ice growth occurs at this time primarily be-
cause there is more initial open water on which frazil ice can form.
Figure 7d shows that the standard deviation of December ice concentration (green curve) peaks
around 2070 and is accompanied by a peak in the variability of the thermodynamic tendency (red
curve) of more than double the magnitude of its dynamic tendency (blue curve). A smaller first
peak in thermodynamic tendency occurs in the 2020s, when ice growth in this region increases
due to increased frazil growth as this region's waters become more open on average in October.
This initial peak may be smaller due to the anti-correlation between dynamic and thermodynamic
tendency, which reduces the effect of the latter. The rapid subsequent decline in ice growth oc-
curs as conditions become too warm for ice growth over much of the October–December period
in the 2050s and 2060s (Fig. 7c). This is reflected in the peak in variability of the thermodynamic
tendency (red curve) approximately corresponding to the timing of the peak in the ice area varia-
bility (green curve) in 2070 (Fig. 7d). The coincidence in their peak variability is similar to that
in Figure 7b and underscores the dominance of thermodynamics over dynamics in regulating the
variability of ice area.
**4. Discussion and Conclusions**

This study has assessed the behavior of interannual Arctic sea ice area variability in the
past and future, using a large set of independent realizations from the CESM-LE and simulations
from 12 models participating in CMIP5.  The results demonstrate the complex, time-varying re-
sponse of the ice pack as it transitions from a relatively stable state during the 20th century to a
more volatile one.  A few of our most important findings are summarized below.
1)      Inter-annual variability of Arctic sea ice cover increases (at least transiently) in all
months in the future as sea ice area and thickness decline, but there is a strong seasonal depend-
ence. There is also a strong seasonal dependence of the magnitude of the maximum ice area vari-
ability in the future, with the greatest magnitude occurring during autumn and winter and small-
est during spring by the time the simulation ends in 2100 (Fig. 2-3). The future peak in variabil-
ity emerges soonest in late-summer months and latest during spring months, and the magnitude
of this peak is positively correlated with the rate of ice loss in every month.
It is possible that the seasonal differences in ice area variability are partially a construction
of the geography of the Arctic Basin, as evident in Fig. 6: when the ice margin is geographically
constrained and unable to expand and contract due to a coastline early in the simulation, there is
a smaller area subject to high ice variability. This explanation was offered by Goosse et al.
(2009) for the same relationship in summer ice area variability, as well as by Eisenman (2010) to
explain retreat rate differences between summer and winter.  In the future, the ice in the central
Arctic Ocean becomes thin enough to expand and contract extensively each season, leading to an
increase in variability. Therefore, variability could be considered to be limited particularly in the
first phase of its time series (Fig. 2) by the inability of ice to spread across a large open area.
Support for this interpretation comes from our calculation of Eisenman's equivalent ice area ap-
plied to Fig. 1 (not shown), which resulted in the largest absolute decline in sea ice during the
winter-spring months, though summer-autumn ice loss was still greater in relative terms. While
useful for approximating potential sea ice extent in the absence of geographic constraints, equiv-
alent ice area is still a theoretical construct; our purpose is to assess the variability of ice cover
that actually exists.   Furthermore, results from Fig. 4 and Fig. 5 suggest that the amount of thin
ice alone can explain the evolution of ice variability in every month, though differences in the
optimal ice thickness by month may require a partial geographical explanation in addition to one
incorporating the components of the thermodynamic tendency of ice area from Fig. 7.
2)      Ice needs to be sufficiently thin before areal variability maximizes, and in CESM-LE the
optimal thickness range is generally between 0.2 m to 0.6 m but with some seasonal dependence
resulting from the ice melt or ice growth processes that dominate in a given season (Fig. 4-5).
The mean ice thickness in late summer and autumn is close to 0.6 m when ice area variability is
highest, but is 0.2 m or less for a grid cell average in the winter.
Increased ice area variability in summer and fall is partly attributable to a higher efficien-
cy of open water formation with the thinning sea ice (Holland et al., 2006) and the fact that
smaller heating anomalies are required to completely melt through vast areas of the thin ice pack
(Bitz and Roe, 2004). We find that the total area of thin ice between the range 0.2 m to 0.6 m is
closely related to how soon and how strongly the peak variability in basin-wide ice area emerges,
and this is primarily a function of variability in ice area's thermodynamic tendency. This is con-
sistent with a physical understanding of this relationship, since ice that is too thin tends to be sea-
sonal and melt off every year, whereas thick ice is more likely to survive the melt season. Sea-

sonal forecasting of September sea ice coverage takes advantage of this concept, with the fore-cast skill improved when initializing ice thickness up to 8 months in advance (Chevallier et al., 2012; Day et al., 2014).

In contrast, ice area variability in November-January arises primarily from inter-annual variability in ice growth (as represented by December in Fig. 7c,d), which is dependent on existing open water conditions and temperature anomalies. The peak in ice area variability in these months also coincides with a slightly lower mean ice thickness of 0.2 m, though it is unclear whether that is due to these ice growth rather than melt processes at work during the winter.

3)   Interannual variability in ice concentration is driven primarily by thermodynamic mechanisms, which are primarily comprised of either ice growth or ice melt depending on the season. Despite being opposing processes, their magnitudes exceed those of dynamic ice processes (Fig. 7).

The thermodynamic tendency in ice concentration is of much greater magnitude than its dynamic counterpart at both the end of the melt season and start of the growth season, and the maximum interannual variability of the thermodynamic term is mostly in phase with that of ice concentration. The inverse relationship between ice area's interannual variability and its interannual rate of change (Figs. 1 and 2) is also found between the thermodynamic tendency and its rate of change (not shown, but inferred from Fig. 7). This is further evidence that ice area variability is primarily driven by thermodynamic processes in the icepack.

The dominance of the thermodynamic tendency is unsurprising and has been established as the relatively more important set of processes controlling sea ice variability, primarily via transport of mid-latitude eddy heat flux anomalies (Kelleher and Screen, 2018), anticyclone passage (Wernli and Lukas, 2018), and increased ocean heat transport (Li et al., 2018). However, the dynamic contribution to changes in ice concentration can likely be substantial in the absence of regional and monthly averaging, and numerous mechanisms have been described that can generate increased ice transport. Recent examples include divergent ice drift events connected to anomalous circulation patterns (Zhao et al., 2018) as well as the collapse of the Beaufort High (Petty, 2018; Moore et al., 2018), both of which may become more common in the future due to preconditioning of the icepack and further intrusion of mid-latitude cyclones into the Arctic.

This study offers a unique contribution by focusing on the projected transient evolution of Arctic sea ice variability throughout the year, as characterized by its response to external greenhouse forcing superimposed on short-term internal variability. A recent study (Olonscheck and Notz, 2017) also identified an overall increase in projected variability of summertime sea ice area in CMIP5, but this conclusion was not consistent across all models, possibly because the analysis did not incorporate the pronounced changes in variability over time as the ice pack diminishes.   Increased inter-annual variability in the CESM Large Ensemble as sea ice declines most rapidly is an important result that needs to be accounted for as the ice-free season expands and the timing of maximum variability shifts from September. We also confirm that this relationship is maintained across CMIP5 models, suggesting that the responsible mechanisms reported here may apply more generally. These results have important implications for marine navigation going forward, indicating that the otherwise auspicious transition to diminished sea ice in every

month may be accompanied by a confounding increase in inter-annual variability of the ice cover
before the ice disappears completely.

**Acknowledgements**

We thank two anonymous reviewers for their helpful comments. Support was provided by the
NOAA Climate Program Office under Climate Variability and Predictability Program grant
NA15OAR4310166. This project is partially funded by the Joint Institute for the Study of the
Atmosphere and Ocean (JISAO) under NOAA Cooperative Agreement NA10OAR4320148,
contribution number 2017-087, the Pacific Marine Environmental Laboratory contribution num-
ber 4671. We would like to acknowledge high-performance computing support from Yellow-
stone (ark:/85065/d7wd3xhc) provided by NCAR's Computational and Information Systems La-
boratory, sponsored by the National Science Foundation.

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

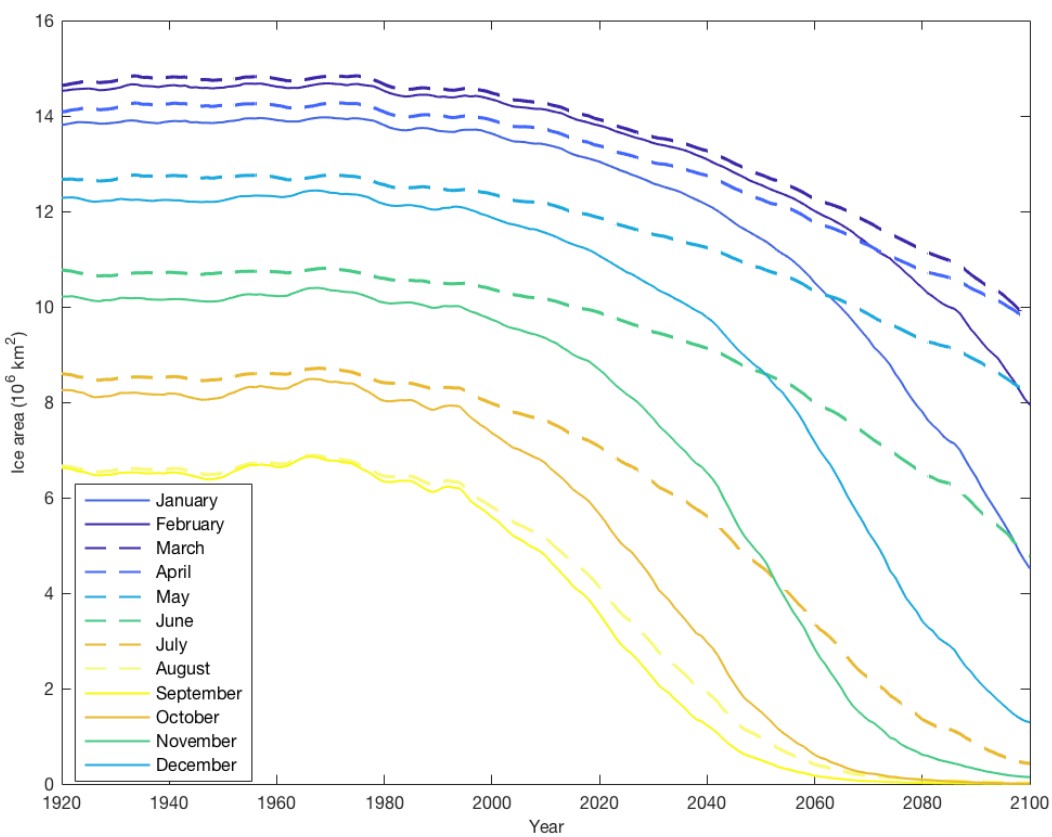

**Figure 1:** The CESM-LE ensemble mean time series of monthly sea ice area ($km^2$ x $10^6$).

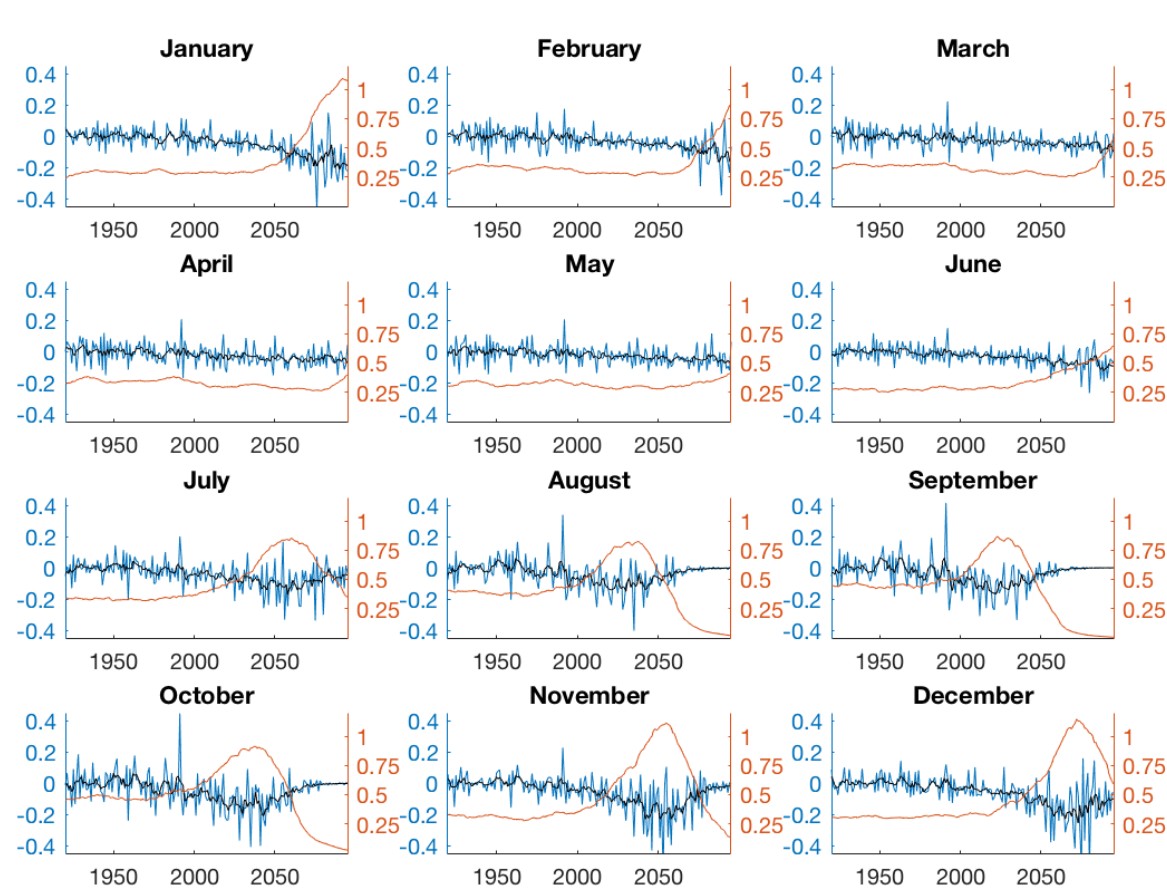

**Figure 2:** The CESM-LE ensemble mean of the 1-year differences in sea ice area (blue; million km$^2$) with their 5-year running mean overlaid (black) and the running standard deviation of the interannual change in sea ice area (gold; million km$^2$).

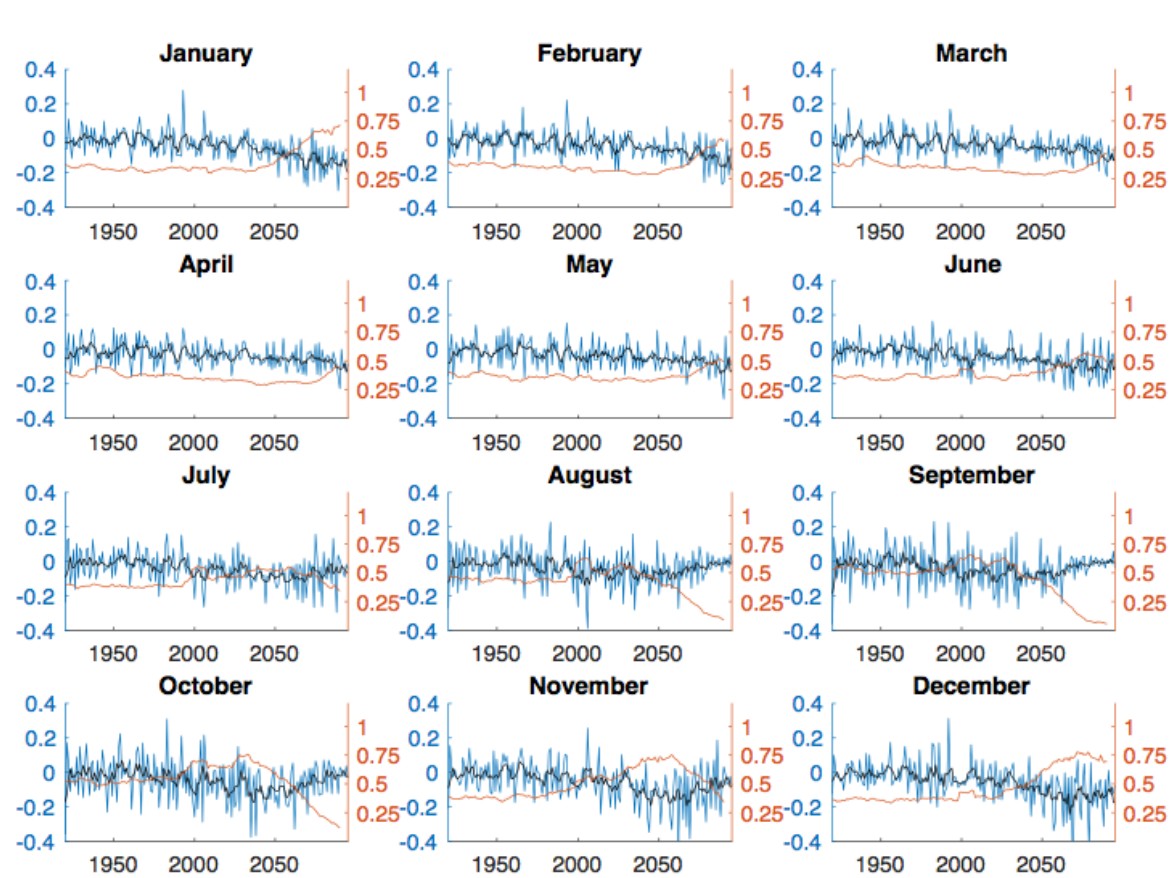

**Figure 3:** As in Fig. 2, but for the ensemble mean from 12 CMIP5 models' sea ice area.

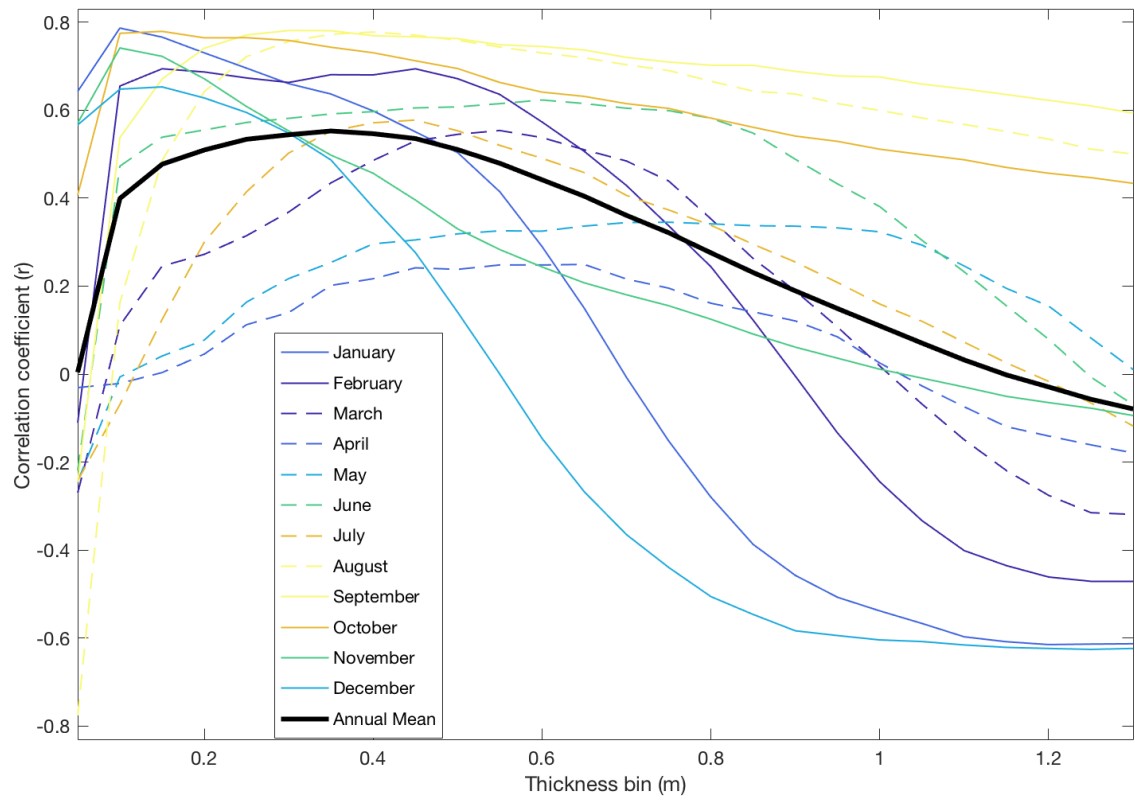

**Figure 4:** Monthly correlation coefficient (r) of the 2000-2100 10-year running standard deviation of 1-year difference in sea ice area with mean grid cell ice thickness binned every 0.05 m of thickness.


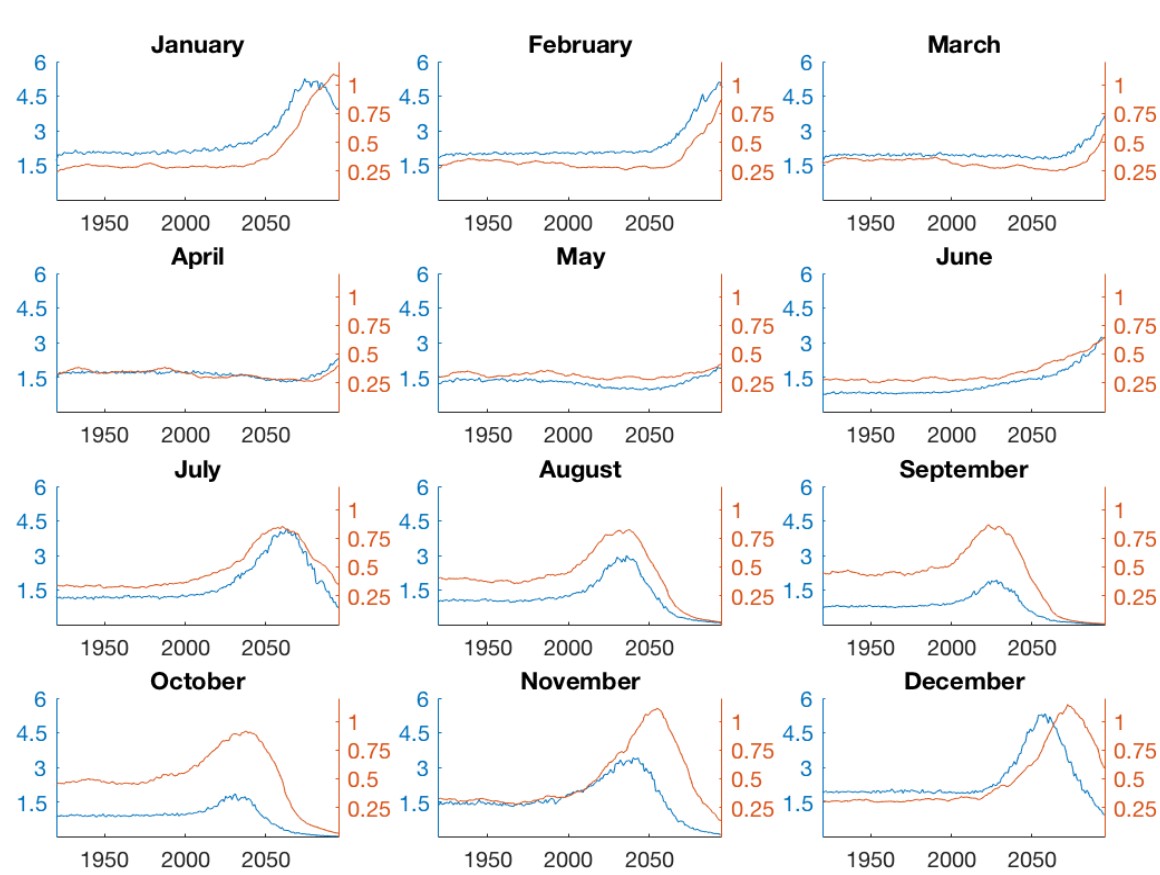

**Figure 5:** The CESM-LE ensemble mean of the 10-year running standard deviation of 1-year difference in sea ice area from Figure 1 (gold; million km$^2$) and the ensemble mean total area of grid cells with mean ice thickness between 0.2 m and 0.6 m (blue; million km$^2$).



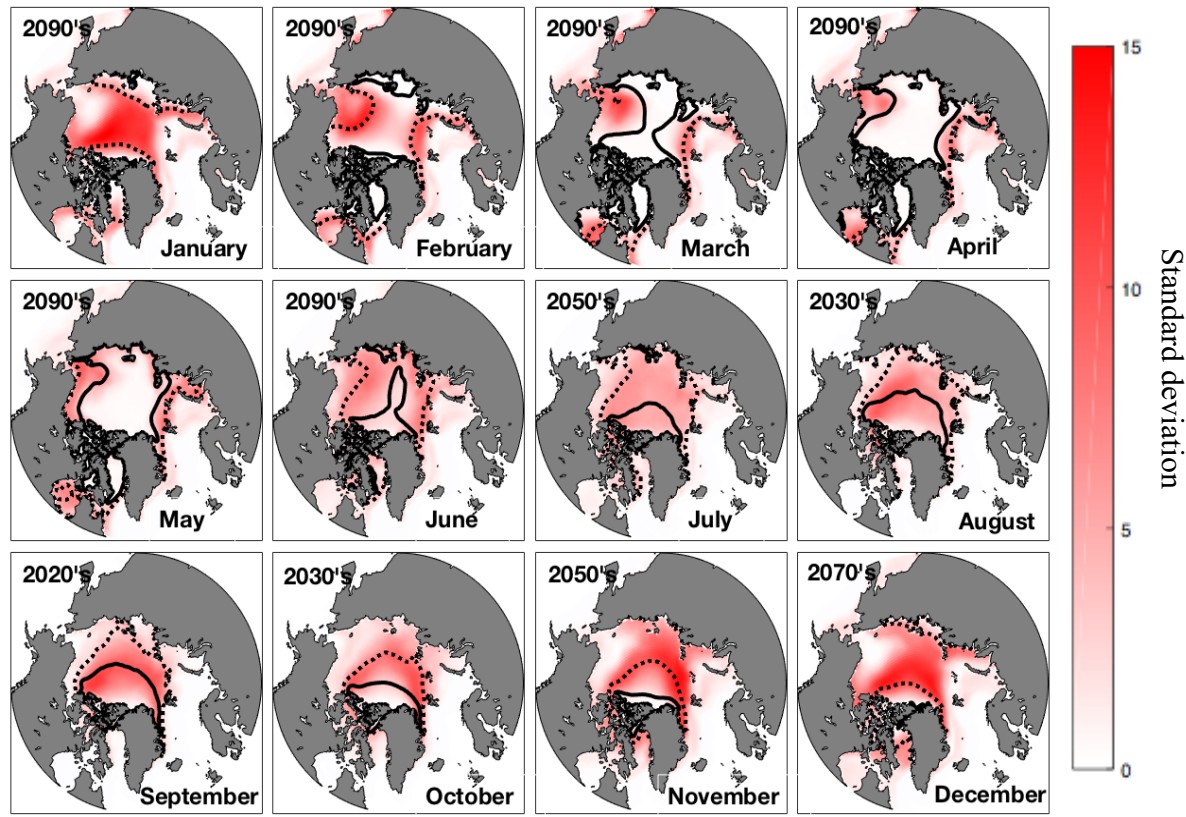

**Figure 6:** Monthly ensemble average in CESM-LE of the 10-year running standard deviation of
ice concentration (%) in the decade when ice area variability is maximum. Mean 0.2 m and 0.6 m
ice thicknesses are indicated by the dotted and solid contours, respectively.



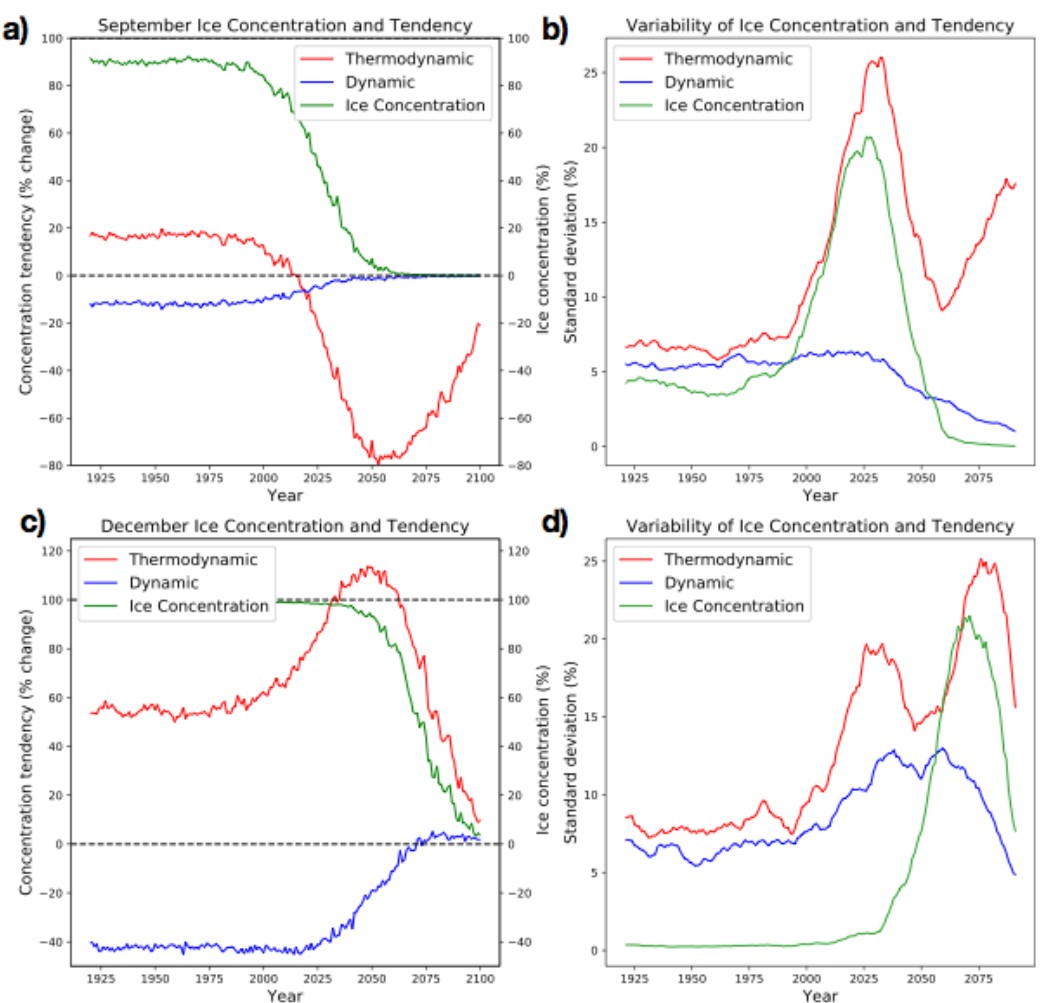

**Figure 7:** Time series of ensemble-mean a) September ice concentration (%) and July-
September averaged concentration tendency (% day$^{-1}$) from dynamics and thermodynamics, and
b) the 10-year running standard deviation of: the inter-annual difference in ice concentration (%),
and July-September ice concentration tendency from dynamics and thermodynamics (% day$^{-1}$).
The same information is presented in c) and d) for December concentration and October-
December ice concentration tendency terms.

# Past and future interannual variability of Arctic sea ice in coupled climate models

John R. Mioduszewski[1], Stephen Vavrus[1], Muyin Wang[2,3], Marika Holland[4], and Laura Landrum[4]

[1]Nelson Institute Center for Climatic Research, University of Wisconsin—Madison, Madison, Wisconsin.

[2]Joint Institute for the Study of the Atmosphere and Oceans, University of Washington, Seattle, Washington.

[3]Pacific Marine Environmental Laboratory, National Oceanic and Atmospheric Administration, Seattle, Washington.

[4]National Center for Atmospheric Research, Boulder, Colorado.

*Corresponding author:* Steve Vavrus, sjvavrus@wisc.edu

## Abstract

The diminishing Arctic sea ice pack has been widely studied, but mostly focused on time-mean changes in sea ice rather than on short-term variations that also have important physical and societal consequences. In this study we test the hypothesis that future interannual Arctic sea ice area variability will increase by utilizing 40 independent simulations from the Community Earth System Model's Large Ensemble (CESM-LE) for the 1920-2100 period, and augment this with simulations from 12 models participating in the Coupled Model Intercomparison Project Phase 5 (CMIP5). Both CESM-LE and CMIP5 models project that ice area variability will indeed grow substantially, but not monotonically in every month. There is also a strong seasonal dependence in the magnitude and timing of future variability increases that is robust among CESM ensemble members. The variability in every month is directly correlated with the average ice retreat rate before there is an eventual disappearance in both terms as the ice pack becomes seasonal in summer and autumn by late century. The peak in variability correlates best with the total area of ice between 0.2 - 0.6 m monthly thickness, indicating that substantial future thinning of the ice pack is required before variability maximizes. Within this range, the most favorable thickness for high areal variability depends on the season, especially whether ice growth or ice retreat processes dominate. Thermodynamic melting (top, bottom, lateral) and growth (frazil, congelation) processes are more important than dynamical mechanisms, namely ice export and ridging, in controlling ice area variability.

**Moved up [2]:** *Corresponding author:* Steve Vavrus, sjvavrus@wisc.edu

*Corresponding author:* Steve Vavrus, sjvavrus@wisc.edu

**Deleted:** ¶

**Deleted:** a set of

**Deleted:** ,

**Deleted:** all

**Deleted:** s,

**Deleted:** and with

**Deleted:** inversely

**Deleted:** m and

**Deleted:** primarily due to

**Deleted:** found to be

# 1. Introduction

Arctic sea ice extent has declined by more than 40% since 1979 during summer (e.g. Stroeve et al. 2012; Serreze and Stroeve 2015; Comiso et al. 2017), primarily as a consequence of greenhouse gas forcing (Notz and Marotzke 2012) but also internal variability (Ding et al. 2017). While this trend is greatest in summer, substantial losses are observed throughout the year (Cavalieri and Parkinson 2012) resulting in an ice season duration that is up to 3 months shorter in some regions (Stammerjohn et al. 2012). Reduced ice area is accompanied by a greater fraction of younger ice (Nghiem et al. 2006; Maslanik et al. 2007a, 2011), which reduces the mean thickness of the basin ice pack (Kwok and Rothrock 2009; Kwok et al. 2009; Lang et al. 2017). As a result, the estimated negative trend in sea ice volume (-27.9% per decade) is about twice as large as the trend in sea ice area (-14.2% per decade; Overland and Wang 2013).

Output from many climate models suggests that the Arctic sea ice cover will not retreat in a steady manner, but will likely fluctuate more as it diminishes, punctuated by occasional Rapid Ice Loss Events (RILEs; Holland et al. 2006; Döscher and Koenigk 2013). The overall decline in ice cover is expected to continue (Collins et al. 2013), and the Arctic may become seasonally ice-free within a few decades, depending on emissions pathway (Stroeve et al. 2007; Wang and Overland 2009; 2012; Massonnet et al. 2012; Wang and Overland 2012; Overland and Wang 2013; Jahn et al. 2016; Notz and Stroeve 2016). However, internal variability confounds prediction of this timing (Stocker et al. 2013; Swart et al. 2015; Jahn et al. 2016; Labe et al. 2018), and even the definition of ice-free differs among Arctic stakeholders (Ridley et al. 2016). Nonetheless, navigation through the Arctic has already increased in frequency as a result of this decline (Melia 2016; Eguíluz et al. 2016), and even more trade routes associated with the increased ice-free season are expected by the end of the 21st century (Aksenov et al. 2015; Stephenson and Smith 2013).

As the Arctic sea ice pack thins and retreats, multi-year ice is being lost and there is consequently a larger proportion of seasonal, thin first-year ice (Kwok et al., 2010, Maykut 1978; Holland et al. 2006). Overall thinner ice may result in an ice pack that exhibits greater inter-annual variability (Maslanik et al. 2007b; Goosse et al. 2009; Notz 2009; Kay et al. 2011; Holland and Stroeve 2011; Döscher and Koenigk 2013), at least partially due to enhanced ice growth and melt (Maykut 1978; Holland et al. 2006; Bathiany et al. 2016a). Decreased ice thickness promotes amplification of a positive ice-albedo feedback, which can magnify sea ice anomalies (Perovich et al. 2007), and thin ice is more vulnerable to anomalous atmospheric forcing and oceanic transport due to the smaller amount of energy required to completely melt the ice (Maslanik et al. 1996, Zhao et al. 2018). For example, pulse-like increases in oceanic heat transport can trigger abrupt ice-loss events in sufficiently thin ice (Woodgate et al. 2012).

Changes in the interannual variability of sea ice have been studied only in a limited capacity, likely because they are only beginning to become visible in September in the present day. Both Goosse et al. (2009) and Swart et al. (2015; their Fig. S6) reported that maximum ice area variability during September occurs once the mean ice extent declines to 3-4 million km². This increased variability may occur due to increased prevalence of RILEs and periods of rapid recovery during this timeframe (Döscher and Koenigk 2013). The thickness distribution during these periods skews toward thinner ice, which is conducive to both rapid ice loss and rapid re-

covery processes (Tietsche et al. 2011; Döscher and Koenigk 2013). Holland et al. (2008) con-
sidered a critical ice thickness that can serve as a precursor to RILEs, but found it more likely
that intrinsic variability played the primary role in the particular RILEs that were studied. More
recently, Massonnet et al. (2018) analyzed the projected variability of sea ice *volume* and its pro-
jected future change in the CMIP5 ensemble, which suggested a monotonic future decrease. The
corresponding variability of sea ice area was investigated by Olonscheck and Notz (2017), but
their analysis was much coarser temporally and seasonally, in that it only compared changes be-
tween entire blocks of time (the historical 1850-2005 period vs. the future 2006-2100 interval)
and was further restricted to the summer and winter seasons.
Building on these previous studies, our paper has two novel aspects. First, we analyze the
transient interannual variability of sea ice area over the course of the year from the early 20$^{th}$
century through the entire 21st century and find very different behavior across the four seasons.
These monthly differences are societally important, because marine access to the Arctic will like-
ly expand beyond late summer as the ice pack shrinks. Second, we detail how interannual sea ice
area variability changes as the ice pack retreats, and we link enhanced future variability to opti-
mal ice thicknesses and to the various thermodynamic and dynamic processes that control ice
area variability. We analyze a large 40-member ensemble from a single GCM, which allows us
to isolate internal variability, which is otherwise muddled with inter-model variability in multi-
model comparisons. This allows us to test the hypothesis that inter-annual Arctic sea ice cover
variability will increase throughout the year in the future as the ice pack diminishes.
**2. Data and Methods**
Ice thickness, concentration, and area were obtained from simulations of the Community
Earth System Model Large Ensemble Project (CESM-LE). Ice concentration refers to the per-
centage of a given grid cell that is covered by ice, while ice area in this study refers specifically
to this percent coverage multiplied by the area of the grid cell yielding a total Arctic ice-covered
area. The CESM-LE was designed to enable an assessment of projected change in the climate
system while incorporating a wide range of internal climate variability (Kay et al. 2015). It con-
sists of 40 ensemble members simulating the period 1920-2100 under historical and projected
(RCP8.5 emissions scenario only) external forcing. The ensemble members are produced by in-
troducing a small, random round-off level difference in the initial air temperature field for each
member. This then generates a consequent ensemble spread that is purely due to simulated inter-
nal climate variability. A full description of the CESM-LE is given in Kay et al. (2015), and sim-
ilar ensembles using the weaker RCP4.5 and RCP2.6 scenarios can be found in Sanderson et al.
209 (2017, 2018).
Another data set used in the current study is the model simulations from the Coupled
Model Intercomparison Project Phase 5 (CMIP5). Although more than 40 models submitted their
simulation results to the Program for Climate Model Diagnosis and Intercomparison (PCMDI),
only 12 of them simulated the Arctic sea ice extent both of the monthly means (each individual
month) and the magnitude of the seasonal cycle (March minus September sea-ice extent) within
20-percent error when compared with observations (Wang and Overland, 2012, Wang and Over-
land 2015). Therefore, we used only these 12 models identified by Wang and Overland (2015) in

**Deleted:** that have focused primarily on variability at the summer ice minimum

**Deleted:** through

**Deleted:** the

**Deleted:** for our study

**Deleted:** has the advantage of

**Deleted:** allowing us to robustly characterize

this study: ACCESS1.0, ACCESS1.3, CCSM4, CESM1(CAM5.1), EC-EARTH, HadGEM2-AO, HadGEM2-CC, HadGEM2-ES, MIROC-ESM, MIROC-ESM-CHEM, MPI-ESM-LR, and MPI-ESM-MR. Among the 12 models, half of them use the same sea ice model as CESM-LE (CICE, Hunke and Lipscomb 2010) or a variation of it. If a GCM provided multiple ensemble members, we only kept up to 5 realizations, so that the total ensemble numbers is close to that used in CESM-LE. There are a total of 33 ensemble members from these 12 models in the RCP8.5 emissions scenario. Sea ice area, rather than ice extent, is computed from these 12 CMIP5 models to be consistent with CESM-LE results.

One of our primary analysis datasets is the time series of monthly ice variables. The ensemble mean of all variables is taken after the statistics are calculated for each ensemble member. 1-year differences in ice area are calculated for each month separately to remove the confounding effect of amplified variability resulting from a downward trend. Finally, a 10-year running standard deviation is applied to the time series of 1-year differences in monthly ice area, centered on a given year. Ten years was chosen to quantify variability over decadal-scale intervals and to provide an adequate number of years for a standard deviation calculation. The timing and magnitude of variability is generally insensitive to the standard deviation window, however, and whether the 1-year difference in ice area or its raw time series is used.

## 3. Results

### 3.1  Sea ice area and its variability

Sea ice area in the CESM-LE is projected to decline in all months in the 21$^{st}$ century, proceeding in three phases: a fairly stable regime of extensive coverage in the 20$^{th}$ century, then a decline, followed by virtually no ice remaining in summer and autumn months (Fig. 1). Sea ice area variability follows an analogous three-phase progression in months spanning mid-summer to early winter (Fig. 2). For example, in September this includes a period of modest variability during the 20$^{th}$ century, then a distinct variability peak in the late 2020s and 2030s that coincides with the maximum rate of ice retreat, and finally negligible variability in the late 21$^{st}$ century as the Arctic reaches near ice-free conditions (Fig. 2). The first two phases of this progression in variability occur for months in late winter to early summer (January-June), and suppressed variability would likely emerge beyond the end of the century, assuming that ice cover in these months would continue to retreat. The maximum rate of ice retreat (negative values of the derivative) occurs at a different time in the 21$^{st}$ century in each month, occurring presently in September but not until the end of the century in spring.

The same relationship between ice area and its variability is maintained across CMIP5 models, though with more noise resulting from the aggregation of many different models rather than ensemble members from a single model (Fig. 3). This is most notable in the sea ice area (1-year difference) time series (Fig. 3, blue), indicating that there is considerable spread in when and how the downward trend proceeds each month, as found in Massonnet et al. (2012), but good agreement that variability increases in this timeframe.

The analysis of ice area variability in Fig. 2 and Fig. 3 follows that of Goosse et al.
(2009) and Swart et al. (2015), but we extend their findings for September to all months and con-
firm that the variability in ice area is maximized as its total basin area decline is well underway
in both CESM-LE ensembles and across CMIP5 models. A direct relationship between the rate
of sea ice retreat and the magnitude of variability is present across all months in CESM-LE and
CMIP5: the standard deviation is highest when ice declines the fastest (Figs. 1 and 2). Further-
more, the magnitude and timing of peak ice area variability in both sets of experiments differs
greatly by season. The peak in magnitude in CESM-LE is most pronounced from November-
January when the running standard deviation of ice area exceeds $1 \times 10^6$ km$^2$, while the lowest
magnitudes occur in April and May, when the downward trend in ice area does not peak prior to
2100 (Fig. 2). Near the end of the 21$^{st}$ century, the running standard deviation also shows an in-
crease in the CMIP5 ensembles from December to June (Fig. 3), very similar behavior to that
displayed by CESM-LE. However the magnitude of the increase in the running standard devia-
tion in the CMIP5 ensemble mean is smaller than that in CESM-LE. This is not surprising, as the
timing of ice retreat varies among models, so averaging them will smooth out the possible sig-
nals. The CMIP5 models therefore provide additional evidence that increased variability is asso-
ciated with decreasing sea ice cover.

## 3.2 Relationship between ice area variability and thickness

Because increasing future concentrations of thin ice are likely a primary factor in in-
creased ice area variability, we next consider the relationship between ice thickness and ice area
variability in CESM-LE. This is done by correlating the standard deviation of basin-wide ice ar-
ea (Fig. 2) with the total area of grid cells with mean ice thickness within a given range for an
aggregation of all years and ensemble members, binned at 0.05 m intervals (Fig. 4). 20th century
data are omitted because both variables are largely stationary for this period. There is a large dif-
ference in the maximum correlation coefficient across seasons, but in most months it peaks be-
tween r = 0.6 and r = 0.8. This peak is associated with the thinnest ice of 0.1 m to 0.2 m from
October to January. There is a broad peak in the correlation coefficient between 0.25 m and 0.40
m in August and September, while July peaks near 0.45 m thickness but with a weaker maximum
correlation coefficient (r = 0.6). In June, r = 0.6 for most ice thicknesses below 0.8 m, and there
is only a weak correlation between these variables in April and May.
The analysis in Fig. 4 allows us to identify a common range of ice thicknesses when ice
area variability generally peaks regardless of the month, which we approximate as 0.2 m to 0.6
m. We next track the temporal evolution of this thin ice throughout the basin by calculating the
total area of ice that falls within that range. The time-transgressive nature of when the peak in
thin ice cover occurs (earliest in September, latest in winter-spring) is consistent with the corre-
sponding timing of the peak future sea ice area variability, suggesting that the emergence of a
sufficiently thin and contracted ice pack is a primary factor for enhanced ice cover variability
(Fig. 5). Both curves match each other in shape, with a steady state early, increasing to a peak
and dropping to zero as the Arctic becomes ice-free. The exception is in the spring and early
summer when neither increases until the end of the 21st century, when ice begins to decline more
rapidly. The two curves are largely in phase as well, with one preceding the other by no more
than 10-20 years in July, August, and November–January. The phase difference is due to the

January

April

July

October

chosen range of ice thicknesses, since the best relationship varies by month (Fig. 4). The two curves are in phase from August-October (Fig. 5) when the 0.2 m to 0.6 m range approximates the best relationship between thickness and variability (Fig. 4). However, ice area variability maximizes after the peak in 0.2 m – 0.6 m thickness area in November–January, because variability is more highly correlated with ice slightly thinner than 0.2 m in these months (Fig. 4; Fig. 5).

There are also notable seasonal differences in the spatial pattern of variability during the decade when variability in ice concentration peaks in CESM-LE (Fig. 6). The largest fluctuations occur in a horseshoe-shaped pattern across the Arctic Ocean in autumn, but they are restricted to the boundaries of the Atlantic and Pacific Oceans in late winter and spring. The result is a larger area of high variability in the second half of the year and into January. The mean 0.2 m (dotted) and 0.6 m (solid) ice thickness contours are overlaid for reference (Fig. 6). The contours correspond closely to the boundary of maximum variability in ice coverage in most months, which is consistent with results from Fig. 4 and Fig. 5. This demonstrates the first-order relationship between thin ice and the variability of inter-annual ice coverage within a given region.

## 3.3 Ice concentration tendency

The strong relationship between thin ice coverage and high concentration variability occurs primarily due to the differing underlying mechanisms controlling ice concentration variability at a given time, namely whether ice is expanding or retreating. To illustrate this, we chose two months representative of these processes, September and December, to conduct an in-depth analysis of the physical mechanisms involved in the time difference in the two curves in Fig. 5. September is the end of the melt season, and therefore the ice concentration over the entire basin in this month reflects the cumulative impact of melt processes throughout the summer. By contrast, December is a time of ice growth, particularly in the future, and thus the ice concentration in this month is largely regulated by cumulative growth processes during the autumn. Using available model output, we calculate the ice concentration tendency (% day$^{-1}$) from thermodynamics and dynamics in the regions where the decadal standard deviation of ice concentration exceeds 30% within the grid cell (Fig. S1) to evaluate the mean ice budget. These regions of maximum variability in September and December closely match those in Fig. 6, though the magnitude is smaller in Fig. 6 due to the standard deviation being a decadal mean. The daily change in ice concentration is a function of dynamic contributions (ice import/export and ridging), thermodynamic melt processes (the sum of top, basal, and lateral), and thermodynamic growth (frazil and congelation). Because antecedent conditions of the icepack can be an important factor for determining ice concentration in the month of interest, we sum these terms over the preceding months (July-September or October-December) and report the net 3-month change in ice concentration resulting from each component.

The most interannually variable ice cover during September occurs primarily in the 2020s and is centered across the central Arctic (Fig. S1a), though this region displays net ice expansion in July-September in the 20th century (Fig. 7a) due to rapid ice growth in September. Thermodynamic processes dominate over dynamics and are of opposing sign during the 20th century, and thermodynamic processes add an average of 20% to the ice concentration of each grid cell in the region by the end of September, compared with a loss of only 10% from dynamical processes

(Fig. 7a). Ice growth diminishes and melt processes accelerate in the early-mid 21st century
when the melt processes reduce ice concentration by more than 75% and the dynamic processes
essentially disappear with less ice to export (Fig. 7a). After 2060, September ice-free conditions
occur, and the thermodynamic term becomes less negative due to reduced areal coverage of ice
in June and hence less ice area to melt over the summer (Fig. 7a).

Because thermodynamic processes dominate in controlling ice concentration in the fu-
ture, they should also be the first-order forcing explaining future ice concentration variability,
particularly given that the magnitude of the dynamic contribution approaches zero by the 2020s
when ice cover is rapidly diminishing. As shown in Figure 7b, the peak interannual variability in
the thermodynamic term (red curve) is indeed several times larger than peak variability of the
dynamic term (blue curve), and the variability in the thermodynamic term maximizes during the
late 2020s in phase with the variability of the ice concentration (green curve) when the thermo-
dynamic term is declining most rapidly in Figure 7a. The variability likely also reflects the influ-
ence of the surface albedo feedback in amplifying summer ice area variations. There is a second-
ary rise in the variability of the thermodynamic term after 2060 (Figure 7b), coinciding with its
rapid rise toward zero in Figure 7a, but ice coverage by this point is confined to a diminishing
area.

From the 20th century well into the 21st century, ice growth occurs in the October-
December period in a similar region of maximum interannual variability as September, except
slightly equatorward (Fig. S1b). Ice export plays a relatively larger role in the regions of interest
in December than in September (Fig. 7c). However, the thermodynamic tendency is still the
dominant term controlling ice concentration within this region of maximum interannual variabil-
ity, and this term increases in the early-mid 21st century to a total of nearly 120%, some of which
is offset by ice export that contributes to a 40% decrease in mean ice concentration in the 20th
and early 21st centuries (Fig. 7c). The increased net ice growth occurs at this time primarily be-
cause there is more initial open water on which frazil ice can form.

Figure 7d shows that the standard deviation of December ice concentration (green curve)
peaks around 2070 and is accompanied by a peak in the variability of the thermodynamic ten-
dency (red curve) of more than double the magnitude of its dynamic tendency (blue curve). A
smaller first peak in thermodynamic tendency occurs in the 2020s, when ice growth in this re-
gion increases due to increased frazil growth as this region's waters become more open on aver-
age in October. This initial peak may be smaller due to the anti-correlation between dynamic and
thermodynamic tendency, which reduces the effect of the latter. The rapid subsequent decline in
ice growth occurs as conditions become too warm for ice growth over much of the October–
December period in the 2050s and 2060s (Fig. 7c). This is reflected in the peak in variability of
the thermodynamic tendency (red curve) approximately corresponding to the timing of the peak
in the ice area variability (green curve) in 2070 (Fig. 7d). The coincidence in their peak variabil-
ity is similar to that in Figure 7b and underscores the dominance of thermodynamics over dy-
namics in regulating the variability of ice area.


**4. Discussion and Conclusions**

This study has assessed the behavior of interannual Arctic sea ice area variability in the
past and future, using a large set of independent realizations from the CESM-LE and simulations
from 12 models participating in CMIP5.  The results demonstrate the complex, time-varying re-
sponse of the ice pack as it transitions from a relatively stable state during the 20th century to a
more volatile one.  A few of our most important findings are summarized below.
1)    Inter-annual variability of Arctic sea ice cover increases (at least transiently) in all
months in the future as sea ice area and thickness decline, but there is a strong seasonal depend-
ence. There is also a strong seasonal dependence of the magnitude of the maximum ice area vari-
ability in the future, with the greatest magnitude occurring during autumn and winter and small-
est during spring by the time the simulation ends in 2100 (Fig. 2-3). The future peak in variabil-
ity emerges soonest in late-summer months and latest during spring months, and the magnitude
of this peak is positively correlated with the rate of ice loss in every month.
It is possible that the seasonal differences in ice area variability are partially a construction
of the geography of the Arctic Basin, as evident in Fig. 6; when the ice margin is geographically
constrained and unable to expand and contract due to a coastline early in the simulation, there is
a smaller area subject to high ice variability. This explanation was offered by Goosse et al.
(2009) for the same relationship in summer ice area variability, as well as by Eisenman (2010) to
explain retreat rate differences between summer and winter. In the future, the ice in the central
Arctic Ocean becomes thin enough to expand and contract extensively each season, leading to an
increase in variability. Therefore, variability could be considered to be limited particularly in the
first phase of its time series (Fig. 2) by the inability of ice to spread across a large open area.
Support for this interpretation comes from our calculation of Eisenman's equivalent ice area ap-
plied to Fig. 1 (not shown), which resulted in the largest absolute decline in sea ice during the
winter-spring months, though summer-autumn ice loss was still greater in relative terms. While
useful for approximating potential sea ice extent in the absence of geographic constraints, equiv-
alent ice area is still a theoretical construct; our purpose is to assess the variability of ice cover
that actually exists.  Furthermore, results from Fig. 4 and Fig. 5 suggest that the amount of thin
ice alone can explain the evolution of ice variability in every month, though differences in the
optimal ice thickness by month may require a partial geographical explanation in addition to one
incorporating the components of the thermodynamic tendency of ice area from Fig. 7.
2)    Ice needs to be sufficiently thin before areal variability maximizes, and in CESM-LE the
optimal thickness range is generally between 0.2 m to 0.6 m but with some seasonal dependence
resulting from the ice melt or ice growth processes that dominate in a given season (Fig. 4-5).
The mean ice thickness in late summer and autumn is close to 0.6 m when ice area variability is
highest, but is 0.2 m or less for a grid cell average in the winter.
Increased ice area variability in summer and fall is partly attributable to a higher efficien-
cy of open water formation with the thinning sea ice (Holland et al., 2006) and the fact that
smaller heating anomalies are required to completely melt through vast areas of the thin ice pack
(Bitz and Roe, 2004). We find that the total area of thin ice between the range 0.2 m to 0.6 m is
closely related to how soon and how strongly the peak variability in basin-wide ice area emerges,
and this is primarily a function of variability in ice area's thermodynamic tendency. This is con-
sistent with a physical understanding of this relationship, since ice that is too thin tends to be sea-
sonal and melt off every year, whereas thick ice is more likely to survive the melt season. Sea-
sonal forecasting of September sea ice coverage takes advantage of this concept, with the fore-
cast skill improved when initializing ice thickness up to 8 months in advance (Chevallier et al.,
2012; Day et al., 2014).
In contrast, ice area variability in November-January arises primarily from inter-annual
variability in ice growth (as represented by December in Fig. 7c,d), which is dependent on exist-
ing open water conditions and temperature anomalies. The peak in ice area variability in these
months also coincides with a slightly lower mean ice thickness of 0.2 m, though it is unclear
whether that is due to these ice growth rather than melt processes at work during the winter.
3)      Interannual variability in ice concentration is driven primarily by thermodynamic mecha-
nisms, which are primarily comprised of either ice growth or ice melt depending on the season.
Despite being opposing processes, their magnitudes exceed those of dynamic ice processes (Fig.
7).
The thermodynamic tendency in ice concentration is of much greater magnitude than its
dynamic counterpart at both the end of the melt season and start of the growth season, and the
maximum interannual variability of the thermodynamic term is mostly in phase with that of ice
concentration. The inverse relationship between ice area's interannual variability and its interan-
nual rate of change (Figs. 1 and 2) is also found between the thermodynamic tendency and its
rate of change (not shown, but inferred from Fig. 7). This is further evidence that ice area varia-
bility is primarily driven by thermodynamic processes in the icepack.
The dominance of the thermodynamic tendency is unsurprising and has been established as
the relatively more important set of processes controlling sea ice variability, primarily via
transport of mid-latitude eddy heat flux anomalies (Kelleher and Screen, 2018), anticyclone pas-
sage (Wernli and Lukas, 2018), and increased ocean heat transport (Li et al., 2018). However,
the dynamic contribution to changes in ice concentration can likely be substantial in the absence
of regional and monthly averaging, and numerous mechanisms have been described that can
generate increased ice transport. Recent examples include divergent ice drift events connected to
anomalous circulation patterns (Zhao et al., 2018) as well as the collapse of the Beaufort High
(Petty, 2018; Moore et al., 2018), both of which may become more common in the future due to
preconditioning of the icepack and further intrusion of mid-latitude cyclones into the Arctic.
This study offers a unique contribution by focusing on the projected transient evolution
of Arctic sea ice variability throughout the year, as characterized by its response to external
greenhouse forcing superimposed on short-term internal variability. A recent study (Olonscheck
and Notz, 2017) also identified an overall increase in projected variability of summertime sea ice
area in CMIP5, but this conclusion was not consistent across all models, possibly because the
analysis did not incorporate the pronounced changes in variability over time as the ice pack di-
minishes.  Increased inter-annual variability in the CESM Large Ensemble as sea ice declines
most rapidly is an important result that needs to be accounted for as the ice-free season expands
and the timing of maximum variability shifts from September. We also confirm that this relation-
ship is maintained across CMIP5 models, suggesting that the responsible mechanisms reported
here may apply more generally. These results have important implications for marine navigation
going forward, indicating that the otherwise auspicious transition to diminished sea ice in every

Deleted: 6

Deleted: y

Deleted: both exceed in

Deleted: at

Deleted: 6

Deleted: 6

Deleted: predicted

Deleted: are robust

Deleted: suggesting

month may be accompanied by a confounding increase in inter-annual variability of the ice cover
before the ice disappears completely.
**Acknowledgements**

Support was provided by the NOAA Climate Program Office under Climate Variability and Pre-
dictability Program grant NA15OAR4310166. This project is partially funded by the Joint Insti-
tute for the Study of the Atmosphere and Ocean (JISAO) under NOAA Cooperative Agreement
NA10OAR4320148, contribution number 2017-087, the Pacific Marine Environmental Labora-
tory contribution number 4671. We would like to acknowledge high-performance computing
support from Yellowstone (ark:/85065/d7wd3xhc) provided by NCAR's Computational and In-
formation Systems Laboratory, sponsored by the National Science Foundation.

**Deleted:** will

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

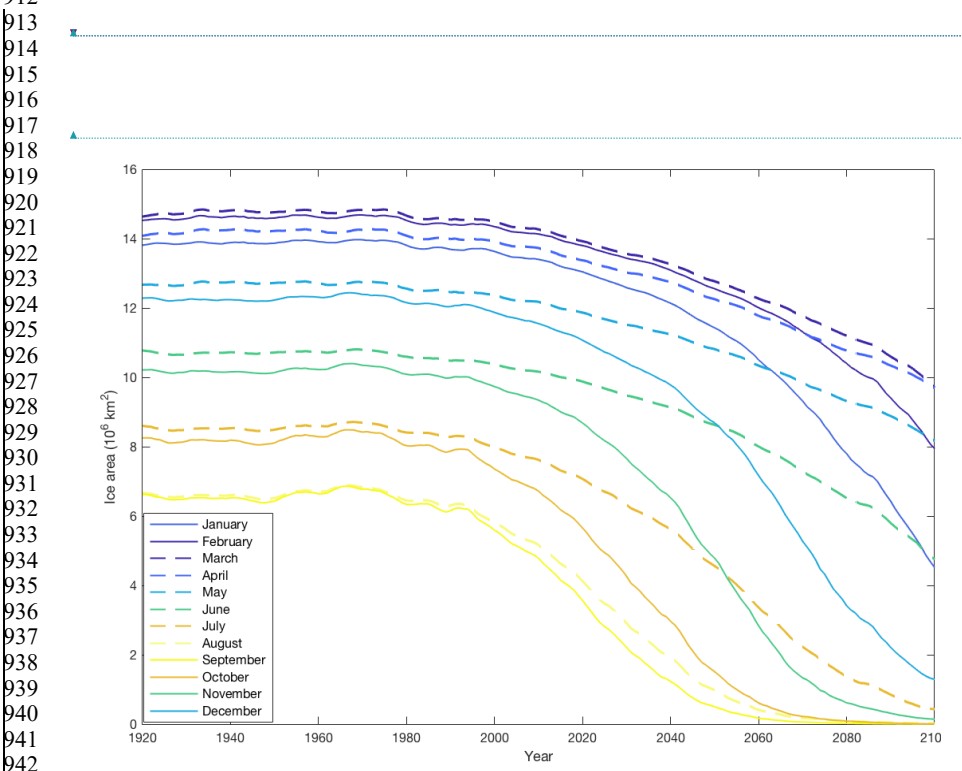

**Figure 1:** The CESM-LE ensemble mean time series of monthly sea ice area (km$^2$ x 10$^6$).

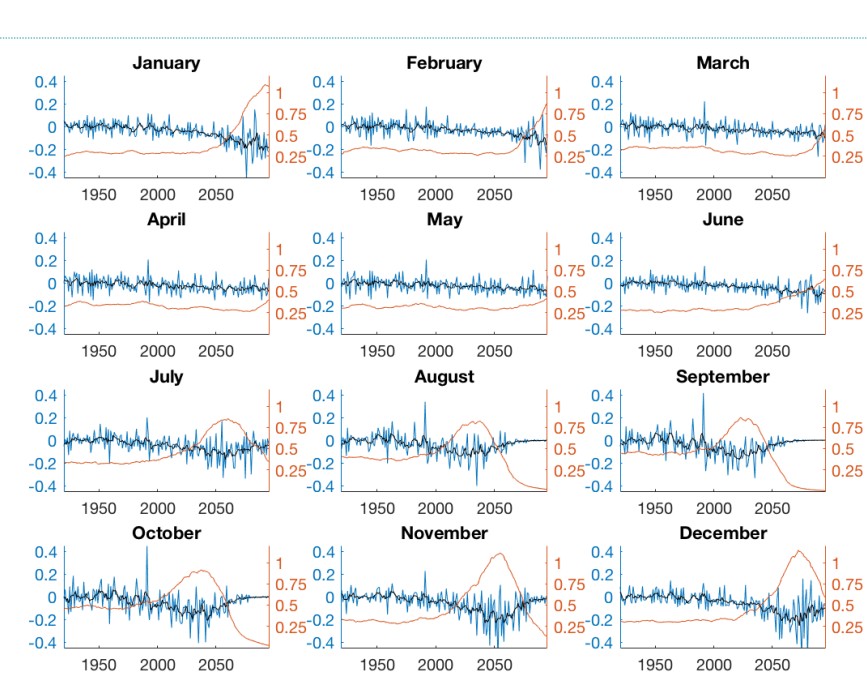

**Figure 2:** The CESM-LE ensemble mean of the 1-year differences in sea ice area (blue; million km$^2$) with their 5-year running mean overlaid (black) and the running standard deviation of the interannual change in sea ice area (gold; million km$^2$).

Formatted: Font: (Default) Times New Roman, 12 pt


**Formatted:** Font: (Default) Times New Roman, 12 pt

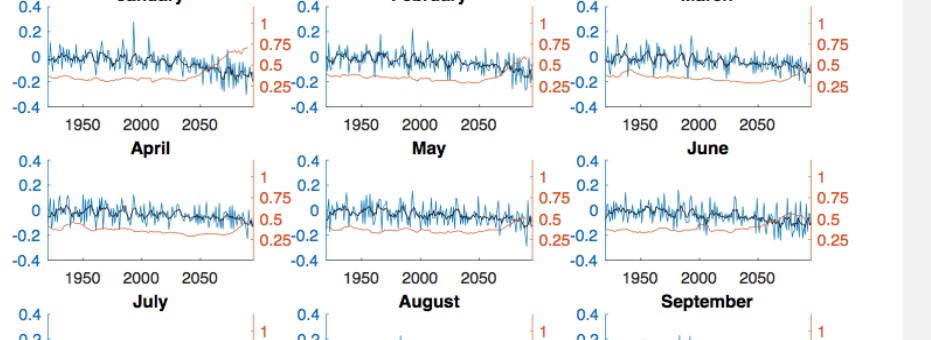

**Figure 3:** As in Fig. 2, but for the ensemble mean from 12 CMIP5 models' sea ice area.

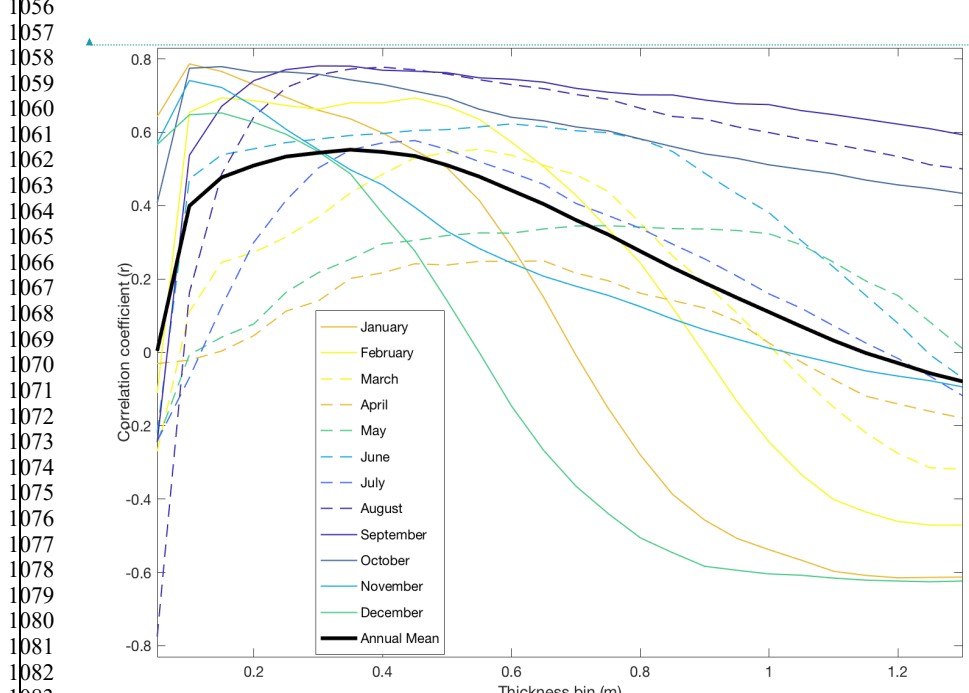

**Formatted:** Font: (Default) Times New Roman, 14 pt, Bold

**Figure 4:** Monthly correlation coefficient (r) of the 2000-2100 10-year running standard deviation of 1-year difference in sea ice area with mean grid cell ice thickness binned every 0.05 m of thickness.

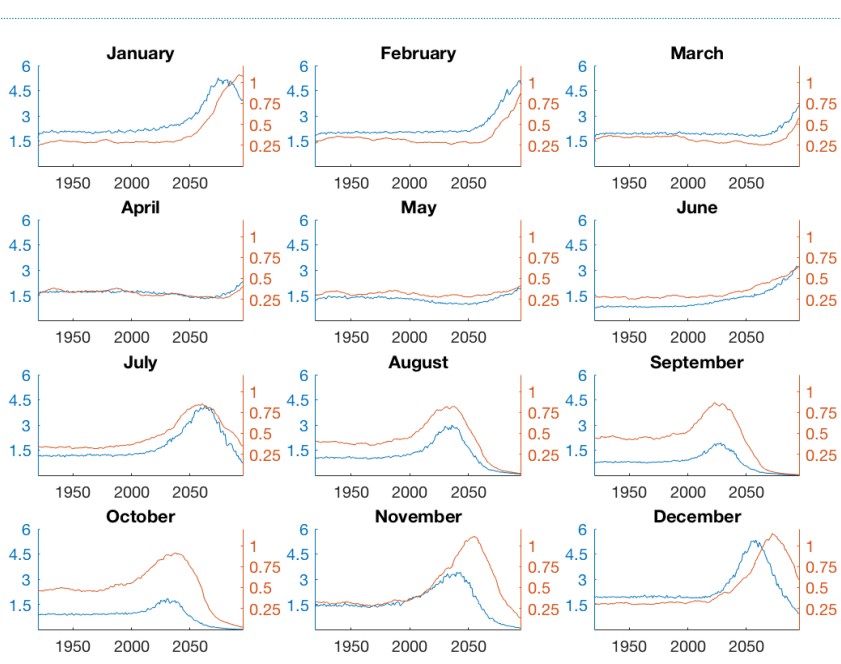

**Figure 5:** The CESM-LE ensemble mean of the 10-year running standard deviation of 1-year difference in sea ice area from Figure 1 (gold; million km$^2$) and the ensemble mean total area of grid cells with mean ice thickness between 0.2 m and 0.6 m (blue; million km$^2$).

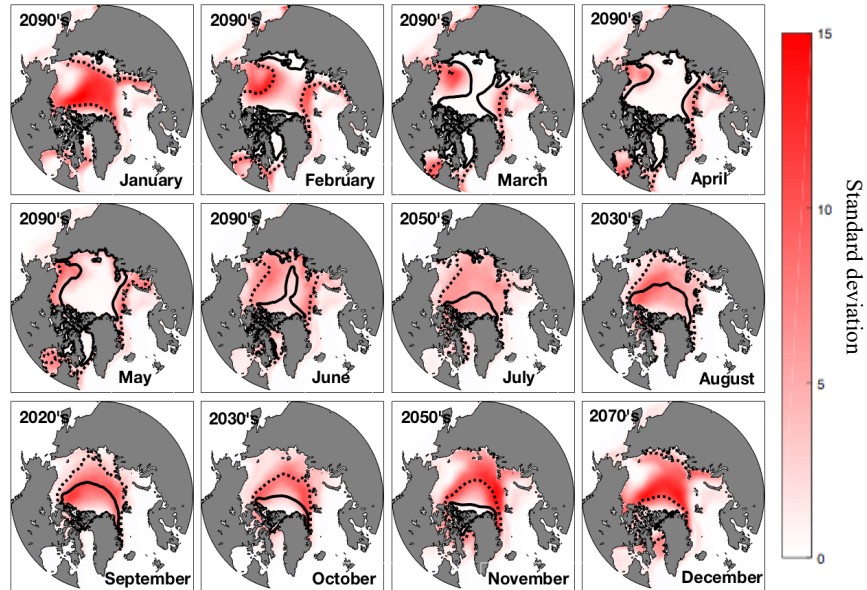

**Figure 6:** Monthly ensemble average in CESM-LE of the 10-year running standard deviation of ice concentration (%) in the decade when ice area variability is maximum. Mean 0.2 m and 0.6 m ice thicknesses are indicated by the dotted and solid contours, respectively.

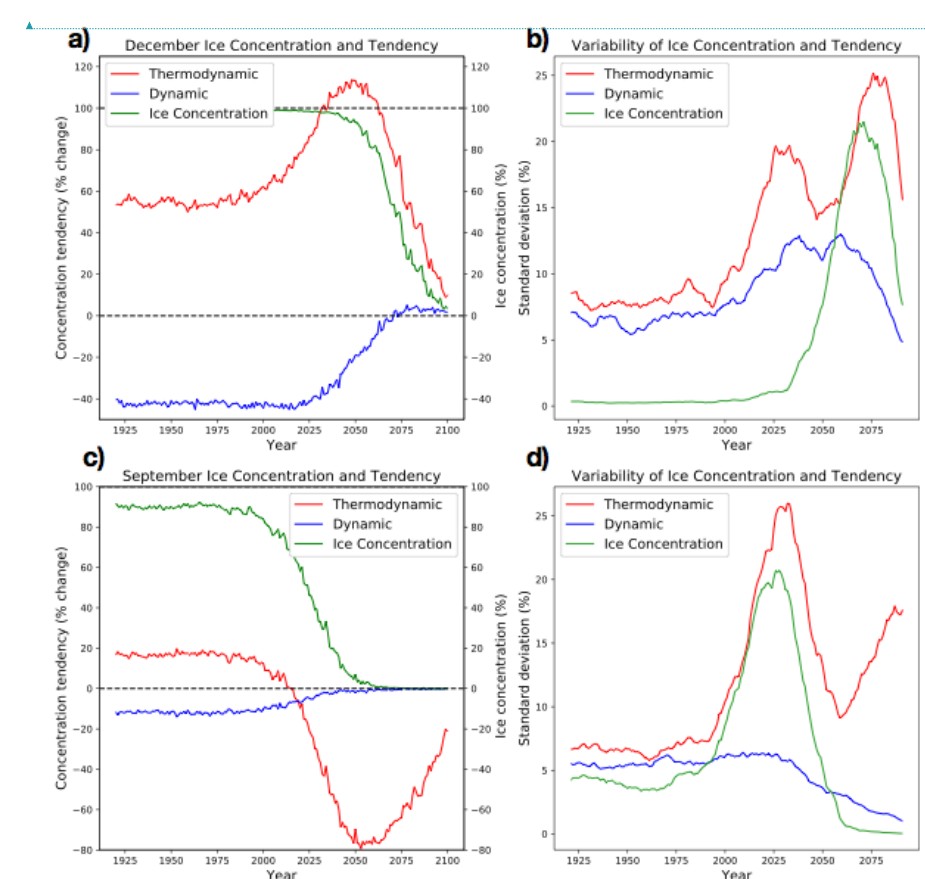

**Figure 7:** Time series of ensemble-mean a) September ice concentration (%) and July-September averaged concentration tendency (% day$^{-1}$) from dynamics and thermodynamics, and b) the 10-year running standard deviation of: the inter-annual difference in ice concentration (%), and July-September ice concentration tendency from dynamics and thermodynamics (% day$^{-1}$). The same information is presented in c) and d) for December concentration and October-December ice concentration tendency terms.

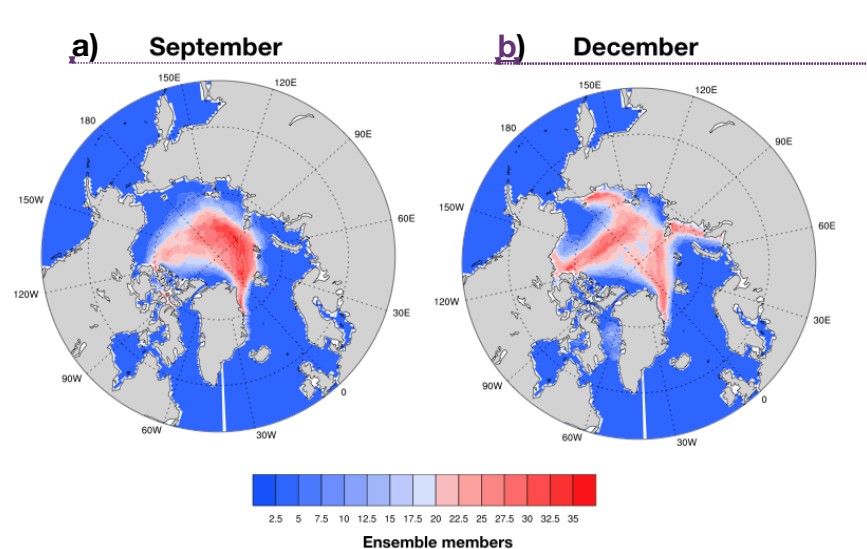

**Figure S1:** Map of the total number of ensemble members where the standard deviation of the 10-year time series of ice area within each grid cell exceeds 30% within a) September's and b) December's decade of maximum variability.

| Page 5: [1] Deleted | Steve Vavrus | 9/3/18 9:52:00 AM |
|---|---|---|
| Page 5: [2] Deleted | Steve Vavrus | 9/3/18 10:15:00 AM |
| Page 6: [3] Deleted | ≥÷÷ | 9/2/18 2:53:00 PM |
| Page 6: [3] Deleted | ≥÷÷ | 9/2/18 2:53:00 PM |
| Page 6: [3] Deleted | ≥÷÷ | 9/2/18 2:53:00 PM |
| Page 6: [4] Deleted | ≥÷÷ | 9/2/18 2:54:00 PM |
| Page 6: [4] Deleted | ≥÷÷ | 9/2/18 2:54:00 PM |
| Page 6: [4] Deleted | ≥÷÷ | 9/2/18 2:54:00 PM |
| Page 6: [5] Deleted | Stephen Vavrus [2] | 8/25/18 9:18:00 AM |
| Page 6: [5] Deleted | Stephen Vavrus [2] | 8/25/18 9:18:00 AM |
| Page 6: [6] Deleted | Stephen Vavrus [2] | 8/25/18 9:20:00 AM |
| Page 6: [6] Deleted | Stephen Vavrus [2] | 8/25/18 9:20:00 AM |
| Page 6: [7] Deleted | Muyin Wang | 8/24/18 12:00:00 AM |
| Page 6: [8] Deleted | Steve Vavrus | 9/3/18 9:57:00 AM |
| Page 6: [9] Deleted | ≥÷÷ | 9/2/18 2:53:00 PM |
| Page 6: [9] Deleted | ≥÷÷ | 9/2/18 2:53:00 PM |
| Page 6: [10] Deleted | Steve Vavrus | 9/3/18 10:01:00 AM |
| Page 6: [11] Deleted | ≥÷÷ | 9/2/18 2:50:00 PM |
| Page 6: [11] Deleted | ≥÷÷ | 9/2/18 2:50:00 PM |