# Peer review of "Past and future interannual variability of Arctic sea ice in coupled climate models John R. Mioduszewski1, Stephen Vavrus1, Muyin Wang2, 3, Marika Holland4, and Laura Lan-drum4 1Nelson Institute Center for Climatic Research, Universit"

_The Cryosphere, 2018_

## Referee Comment (RC1) · Anonymous Referee #1 · 23 Jun 2018

general comments:

This is an interesting study addressing the evolution of the interannual variability of Arctic sea ice area and its causes. The study is based on analyzing historical and RCP8.5 simulations of CESM-LE and, in part, 12 CMIP5 models. The study primarily finds an inverse relationship between the rate of sea ice retreat and the magnitude of variability. The authors further show that a sufficiently thin ice cover fosters the variability and that thermodynamical processes dominate over dynamical processes in causing this variability. Although I find the study relevant, well written and structured, I have some major concerns about the novelty of the findings, some methodological aspects and the robustness of their conclusions. I recommend publication in TC only if these major concerns will be addressed.

[Figure]

specific comments:

Title: The paper does not only address the future interannual variability but also the past.

**1 ll. 47-48 The "important physical and societal consequences" could be given more specific.**

**2 ll. 60-62 It is not clear from the abstract what "thermodynamic processes" exactly mean. I would like to have this more specific (e.g., open water formation efficiency), especially because you specifically name the dynamic processes, which you find to be less important, but not the important thermodynamic ones.**

**3 l. 106 Where does the judging statement "likely" comes from? Is this justified in the given references according to IPCC language. As you mention, the likelihood of summer ice-free conditions strongly depends on the emission scenario. For RCP8.5, it might be rather certain, for RCP2.6 it is not. "Likely" is vague here.**

**4 l. 117 To me, there is no logical link between the reduction in sea-ice extent and the loss of multi-year ice. The reduction in sea-ice extent is not obviously the cause for the loss of multi-year ice. Please rephrase.**

**5 ll. 122-123 How does decreased ice thickness amplifies the ice-albedo feedback? Please explain.**

**6 ll. 141-142 I appreciate that you specifically mention the novel aspects of your study. However, I find these aspects only partly novel. The first aspect is not truly novel. Olonscheck and Notz, 2017 (Consistently Estimating Internal Climate Variability from Climate Model Simulations. Journal of Climate) distinguish changes in the variability of winter and summer Arctic sea ice area. The second aspect is also touched by Olonscheck and Notz, 2017 but your study goes beyond this by investigating the underlying processes for the model-simulated changes in CESM-LE. However, the very recent study by Massonnet et al., 2018 (Arctic sea-ice change tied to its mean state through**

thermodynamic processes. Nature Climate Change) covers parts of your findings. I recommend to more clearly work out the novel aspect of your study, to distinguish your results from the mentioned studies, and to discuss your results in the context of their findings.

**7 ll. 147-148 I don't believe that the internal variability is robustly characterized from just one model. The internal variability largely differs between the CMIP5 models. How do we know that CESM-LE is representative? I assume you mean that 40 ensemble members allow to robustly quantify the internal variability WITHIN THAT MODEL, but I don't believe that your statement is correct as it is now. Please be more precise here.**

**8 ll. 162 A medium ensemble of 15 members for RCP4.5 described in Sanderson et al., 2015 (A new ensemble of GCM simulations to assess avoided impacts in a climate mitigation scenario. Climatic Change) and recent ensembles, e.g., for RCP2.6 described in Sanderson et al., 2017 (Community climate simulations to assess avoided impacts in 1.5 and 2°C futures. Earth System Dynamics) also exist.**

**9 ll. 170-172 For two reasons, I am not convinced by the usefulness of this selection criteria. First, because the threshold of 20-percent error seems arbitrary to me. How is this justified? Second and more importantly, there is no reason to believe that models that fit the observations comparatively well are better than others because of the large influence of internal variability. When taking model-specific internal variability into account the sea-ice simulations of most CMIP5 models are plausible. I would like to see whether or not your basic conclusions change when using the full set of CMIP5 models. Also, as a reader I would like to know which CMIP5 models you used without having to look this up in Wang and Overland, 2015.**

**10 ll. 176-177 To calculate the statistics for each of the 33 ensemble members and to then average them gives a biased estimate, because models with more ensemble members have a larger weight than models with only few (or even one) members. Again, I would like to see whether your basic conclusions would change when you**

always use e.g. three ensemble members from a model. As it is, I don't find the approach convincing.

**11 ll. 212-217 I find the analysis of the CMIP5 models rather weak. To me, it is no proof that the variability is indeed increasing as shown by e.g., Goosse et al. 2009. This is because I see no logic behind simply averaging the CMIP5 models. As you write, the timing of ice retreat is very different in the different models, so averaging them will smooth out possible signals. For instance, one could normalize the timing of sea ice retreat before doing the analysis. I think that more analysis of the robustness of the results based on the CMIP5 models is needed.**

**12 ll. 230-232 Related to the previous comment, I would like to know which of the two reasons is more relevant.**

**13 ll. 219-220 See again Olonscheck and Notz, 2017.**

**14 ll. 304-322 It is not very clear to me how exactly you calculate the thermodynamic and dynamic component. For instance, do you sum up top, basal and lateral melt for the thermodynamic melt component? I think I can guess what you did, but it is not written down precisely.**

**15 l. 373 I recommend one or two introductory sentences here to guide the reader. This would also help to improve the structure of the discussion section.**

**16 l. 443 This should be "projected", instead of "predicted".**

**17 ll. 448-449 I very much appreciate that your work includes the analysis of CMIP5 models. But I question that the presented analysis is sufficiently well done to justify this statement on robustness. Especially, because the CMIP5 models are only used for section 3.1 and not in the later sections that deal with the mechanisms. The questionable (see comment #11) and generally weak inverse relationship between variability and rate of retreat that you show for the CMIP5 models does NOT necessarily imply (and also does not suggest) that the same mechanisms are at work like the ones you**

describe for CESM-LE. This statement is too strong. I recommend to either extend your analysis of the mechanisms to the CMIP5 models (if possible) or further weaken or delete this statement.

technical corrections:

l. 331 I prefer "the variability in the thermodynamic term", rather than "the thermodynamic term variability"

Figures: I suggest to make the figures look more consistently, i.e. Figures 3 and 6 like Figures 1, 2 and 4. Also, I find the different axis labeling in Figure 6a and 6c confusing. For Figure 6, a title for each panel would increase the readability and lines at 0 percent and 100 percent in panels a and c, too.

l. 345 frazil = frazil ice?

References: Comiso et al ... The year of publication is missing. Zhao et al., 2018 ... This reference appears twice.

---

## Referee Comment (RC2) · Anonymous Referee #2 · 27 Jun 2018

This study by Mioduszewski and co-authors is concerned with the future (and to some degree past and present) variability of the Arctic sea ice cover in GCMs. The article focuses on seasonal aspects of the variability in sea ice area, and on potential drivers of such variability. The authors find a strong correlation between ice area variability and ice thickness, and argue that thermodynamic processes have a stronger impact on variability than dynamic processes.

The study is concerned with an important topic that fits well within the scope of The Cryosphere. I agree by-and-large with the comments of the other reviewer, and would hope to see some substantial revisions of the manuscript. Furthermore, several parts of the manuscript are marked by a somewhat disappointing standard of language and presentation, in particular given the experience and seniority of the co-authors. Below

I will detail concerns that I have in addition to those voiced by the other reviewer.

General comments:

1. The abstract and introduction should be thoroughly revised (see specific comments below). The writing improves from Section 2 onward.

2. Please consider the geographic muting effect of Eisenman (2010) in more detail. i.e. what do analogues to Fig S1 and Fig 1 look like when using Eisenman's "equivalent ice extent"? This would help quantify the role that the distribution of land around the Arctic basin plays in this context.

3. I share the concerns of the other reviewer in that the discussion of the CMIP5 analysis is somewhat vague and incomplete. It also should be put more clearly in context with other recent work on the subject.

4. In Sec 3.3 it seems odd to choose Sept and Dec as months to study the roles of ice retreat and expansion, respectively. First, the sea ice minimum occurs typically in mid-Sept, which means that there's substantial ice expansion in the 2nd half of the month (as remarked in L.314). Thus, if the authors want a fully retreating month, why not choose August? December, on the other hand, is fairly early in the ice expansion phase, so if the aim is to capture as much as possible of the preceding expansion, why not choose February? Or January? To that point, in the conclusions (L.413-414) the authors relate Fig 6c,d to "Nov-Jan" variability (rather than to Oct-Dec, as used in the analysis).

5. As pointed out by the other reviewer, this work needs to be put carefully in context with the very recent paper by Massonnet et al ("Arctic sea-ice change tied to its mean state through thermodynamic processes", Nature Climate Change, 2018). I appreciate that the latter study was published after this one was submitted.

Specific comments:

I would suggest moving Fig S1 to the main text as Fig 1. I'd also suggest color-coding

the different months sequentially in this figure so that the seasonal cycle becomes more visible.

L.49: I would suggest deleting "independent".

L.52-54: This sentence is somewhat confusing. Some months see an essentially monotonic increase and it's not immediately clear what part of the sentence refers to CESM-LE and what to CMIP5. I suggest rephrasing and/or splitting into 2 sentences.

L.55: "inversely" correlated. This is used at several points in the ms. Unless I'm mistaken, isn't the rate of retreat "directly" correlated with the variability? In other words, the larger the rate of retreat, the larger the variability (?). At L.428 the authors talk about the rate of change. Here I can see the inverse relation: the more negative the rate of change, the larger the variability.

L.58: "...indicating that [for most of the years (?)] substantial future thinning ..."

L.59-60 "... depends on the season, primarily due to whether ..." This could be written more clearly.

L.98 "... reduces the [mean] thickness of the basin ice back ..."

L.100 " ... the [estimated] negative trend ..."

L.103 "[Output from] many climate models suggest[s] ..."

L.113-114 rephrase

L.117 I agree with the other reviewer that the implied causality between reduced extent and loss of multi-year ice is misleading.

L.118 "Increased thin ice ...". Replace with "Overall thinner ice ..."

L.121 "... ice growth and retreat rates ..." I'd argue this should either be "expansion and retreat" or "growth and melt"

L.129 "relationship between ice area and its variability". Do the authors mean the

"mean ice area" and the "variability in ice area"?

L.130 "... it is only beginning to become visible ..." The relationship is becoming visible? Does this mean that we are starting to observe a correlation between the mean ice area and the variability of ice area? Please clarify.

L.143 "... monthly differences are [societally/economically?] important ..."

L.148 "... characterize internal variability [of CESM]" (see other reviewer's comment)

L.198 "... follows [an analogous] three-phase progression ..."

L.222 "inverse" see comment at L.55

Fig.3 I find it hard to decipher the individual curves here. What about splitting the figure into 2 panels, with panel (a) showing spring/summer months and (b) showing fall/winter months. The missing curves in each panel could shown as faint gray in the background for reference. Again, I would use a sequential color map.

L.240 "... between ice thickness and [ice area] variability ... " Otherwise it might be read as "ice thickness and ice thickness variability"

L.267-272 Would it be worth showing another thickness curve (<0.2 m) in Fig 4 to illustrate the phase dependence (and different area coverage) for different ice thicknesses?

Fig 5: The left hand side of the bounding boxes was cut off. Also, the resolution of the figure was low (jpg? Better to use png with resolution > 150 dpi). It'd be nice to add the respective decade in the top left corner of each panel.

L.275: Please mention CESM-LE in the caption.

L.287: "... thin ice and [the variability of] inter-annual ice coverage ... "

L.280-282: How much of this difference is simply due to the limited run length of the simulations? In other words, once the ice retreats further in winter and spring after 2100, would one then also see the horse-shoe pattern in those seasons? Conversely,

in the summer, are the regions of high variability restricted to the Arctic boundaries during earlier decades in the simulations?

L.293 "expanding" rather than growing

L.304 Why is a different method used here to calculate the standard deviation?

L.337-340 delete "rather than melt". Split into two sentences?

L.342 "[mid] 21st century"

L.345 "frazil [ice]"

Fig 6: The (a)-(d) labels are too big and bold, and the rest of the text in the figure is too small.

L.376 Isn't the smaller magnitude of spring variability just a result of the time series ending in 2100 (before the ice edge retreats into the Arctic basin in spring)?

L.415 "... ice area variability [in winter] also coincides ..."

---

## Author Comment (AC1) · 26 Aug 2018

A full response to both reviews has been attached as a pdf to preserve formatting, with our responses in red. Figures with substantive changes have been attached, along with a figure showing analysis of equivalent ice area as requested. The revised introduction and data/methods section has been included as well due to some of the more extensive edits occurring in response to the reviewers' suggestions.

Please also note the supplement to this comment:
https://www.the-cryosphere-discuss.net/tc-2018-100/tc-2018-100-AC1-supplement.zip

[Figure]

[Figure]

Figure showing Ice area plotted against Year, with monthly lines.

**Fig. 1.** Figure 1

[Figure]

[Figure]

**Fig. 2.** Figure 3

[Figure]

**Fig. 3.** Figure 6

[Figure]

**Fig. 4.** Equivalent Ice Area

---

## Referee Report (RR1)

Thanks for the careful response to the reviewer comments on „Past and future interannual variability of Arctic sea ice in coupled climate models". While the authors mainly explain the approaches used in their initial submission, I would have wished to see some additional research to elaborate on the new aspects of the study, as well as a more elaborated comparison of their results to CMIP5. Although I feel that some chances are missed for a high impact study, I find the detailed response sufficiently conclusive to recommend publication in TC.

I have two minor recommendations that the authors may want to consider before publication:

1. Please be consistent with the terminology by using either „bottom melt" or „basal melt" throughout the manuscript.

2. Please cite the mentioned original studies in addition to Perovich et al (2007). [ll.83-84]

---

## Author Response (AR2)

Dear Editor Notz,

We appreciate your comments regarding the recent resubmission of our article, "Past and future interannual variability of Arctic sea ice in coupled climate models". In this letter, we explain our responses to your three specific points (italicized).

1. I have noted that the submitted manuscript version with tracked changes is not correctly showing all differences between the originally submitted manuscript and the new manuscript. In particular, I stumbled across changes that have been made to the new version that do not show up in the tracked-changed version, while changes show up in the version with tracked changes that are not apparent in the new version. While the differences I was able to spot are minor, they carry the possibility that also major changes are not marked in the version seemingly showing the changes. To prevent this possibility, I ask you to submit a version with tracked changes that shows the true difference between the two versions.

We apologize for this confusion. It turns out that we failed to submit the most updated file containing tracking changes, so some of our revisions were not visible. In this new resubmission, the file containing tracked changes should allow a direct comparison with our original version.

2. I note that very little reference is made and no additional work has been done in the new version of the manuscript regarding the possible lack in novelty that was strongly pointed out by both reviewers. Note that I will reject this manuscript should the reviewers stand by their view that there is too little novel science presented in this work.

We appreciate this perspective, and we are keenly aware that our paper needs to possess sufficient novelty to warrant publication. In our interpretation, the reviewers did not sharply criticize our study for a lack of originality, but rather that the novel aspects we had cited were "only partly novel" (Reviewer 1) and that we need to distinguish our findings from those in the recent related studies by Massonnet et al. and Olonscheck and Notz (both reviewers). We thank both reviewers for bringing these new papers to our attention, since both studies were so recent that we had not included them in our original manuscript.

In this resubmission, we have explicitly cited both of these studies and how our results differ from them. In particular, the Massonet et al. (2018) analysis only addressed sea ice volume variability and its projected future change, which is to be a monotonic decrease, unlike the more complicated time-dependent response of ice area variability that is our focus. Our findings also help resolve the findings in Olonscheck and Notz (2017), whose analysis is of future changes in sea ice variability is much coarser temporally and seasonally (the 2006-2100 time block for JAS and JFM), and less conclusive: (a) "the future internal variability of summer Arctic sea ice area possibly increases. . . ", and (b) ". . . winter and summer Arctic sea ice area show inconsistent model responses" [in CMIP5]. They also identify a decrease in future sea ice volume variability, like Massonet et al., but that is not directly pertinent to our findings on sea ice area variability.

We have added these two new studies to the introductory material, where we highlight previous findings on this topic and distinguish our focus from their work [lines 99-104]. In addition, we now compare and contrast our results with those of Olonscheck and Notz in the final paragraph of the paper to underscore the uniqueness of our study [lines 408-412].

In addition to these sharpened contrasts, we have strengthened the original paragraph at the end of the Introduction, in which we explain the two main novel aspects of our study. To this revised version, we have clarified that our analysis investigates the transient (continuous) variability of sea ice area over the course of the year, all of which are important additions to the literature on this topic, particularly because we find "very different behavior across the four seasons". The last finding is especially relevant as the Arctic ice pack transitions toward increasing open water beyond the late summer.

3. I also note that no reference has been made to the remark of the second reviewer that "several parts of the manuscript are marked by a somewhat disappointing standard of language and presentation, in particular given the experience and seniority of the co-authors." I encourage you to seriously consider this criticism for the revised version of this manuscript.

We thank the reviewer for their constructive criticism and the specific places in the text that they felt needed improving. Although we hadn't explicitly explained how we had addressed this comment in our recent resubmission, our point-by-point response to all of the reviewer's suggestions was meant to imply that. To clear up confusion, though, we have added a new response near the top of our replies to Reviewer 2 regarding the sub-standard language and presentation:

"We have revised the manuscript accordingly, as described in our point-by-point responses below, many of which directly pertain to language and presentation, especially those listed under 'Specific comments'."

We note that we have addressed every such point raised by that reviewer. In most cases we have simply followed their helpful suggestions and listed the corresponding line number in the manuscript where the improved wording can be found.

Thank you again for your assistance in shepherding the review of our manuscript. Please do not hesitate to contact us with any other questions or comments.

Regards,

John Mioduszewski Steve Vavrus Muyin Wang

**Anonymous Referee #1**

**general comments:**

This is an interesting study addressing the evolution of the interannual variability of Arctic sea ice area and its causes. The study is based on analyzing historical and RCP8.5 simulations of CESM-LE and, in part, 12 CMIP5 models. The study primarily finds an inverse relationship between the rate of sea ice retreat and the magnitude of variability. The authors further show that a sufficiently thin ice cover fosters the variability and that thermodynamical processes dominate over dynamical processes in causing this variability. Although I find the study relevant, well written and structured, I have some major concerns about the novelty of the findings, some methodological aspects and the robustness of their conclusions. I recommend publication in TC only if these major concerns will be addressed.

Thank you for the thoughtful evaluation. In this revised version, we have addressed all three of the reviewer's concerns regarding novelty, methodology, and robustness. Our reply to all of these concerns are incorporated into the point-by-point responses below.

**specific comments:**

Title: The paper does not only address the future interannual variability but also the past.

Good point. The title has been changed to "Past and future interannual variability of Arctic sea ice in coupled climate models".

**1 II. 47-48 The "important physical and societal consequences" could be given more specific.**

Although there isn't room in the abstract to explain the physical and societal consequences of short-term variations in sea ice as the pack diminishes, some of these impacts are covered in the Introduction. These include enhanced marine navigation through the Arctic, amplified positive ice-albedo feedback, and increased sea ice sensitivity to ocean heat transport initiated upon short-term declines in ice cover. The components of the surrounding ecosystem will also adjust as the physical environment changes. [Lines 82-87, 109-110]

**2 II. 60-62 It is not clear from the abstract what "thermodynamic processes" exactly mean. I would like to have this more specific (e.g., open water formation efficiency), especially because you specifically name the dynamic processes, which you find to be less important, but not the important thermodynamic ones.**

We have added more specificity by clarifying that the thermodynamic processes involve melting (top, bottom, lateral) and growth (frazil, congelation). There is no more space to get into additional details within the short abstract. [Line 45]

**3 I. 106 Where does the judging statement "likely" comes from? Is this justified in the given references according to IPCC language. As you mention, the likelihood of summer ice-free conditions strongly depends on the emission scenario. For RCP8.5, it might be rather certain, for RCP2.6 it is not. "Likely" is vague here.**

To avoid bogging down in the spread and plausibility of the various RCP scenarios, we have changed the sentence to read that that the Arctic "may" become seasonally ice-free within a few decades. [Lines 66-67]

**4 I. 117 To me, there is no logical link between the reduction in sea-ice extent and the loss of multi-year ice. The reduction in sea-ice extent is not obviously the cause for the loss of multi-year ice. Please rephrase.**

Although the reduction of sea-ice extent does not directly cause a loss of multi-year ice, past studies do show that a large portion of the reduction of sea-ice extent in the past decade is associated with the loss of multi-year ice (Kwok et al., 2010). That reference is now added. We have also rephrased the sentence, "As the Arctic sea ice pack thins and retreats, multi-year ice is being lost and there is consequently a larger proportion of seasonal, first year ice." [Lines 77-79]

**5 II. 122-123 How does decreased ice thickness amplifies the ice-albedo feedback? Please explain.**

As demonstrated by observations (e. g., Grenfell and Maykut 1977) and applied in various parameterizations and models (e. g., Maykut 1982, Ebert and Curry 1993), the albedo of thin, first year sea ice is lower than that of thick, multiyear ice. The CICE sea ice model used in CESM-LE also lowers the surface albedo for thin ice floes (Hunke and Lipscomb 2010).

Grenfell, T. C., and G. A. Maykut, 1977: The optical properties of ice and snow in the Arctic Basin. J. Glaciol., 18, 445-463.

Maykut, G. A., 1982: Large-scale heat exchange and ice production in the central Arctic. J. Geophys. Res., 87, 7971-7984.

Ebert, E. E., and J. A. Curry, 1993: An intermediate one-dimensional thermodynamic sea ice model for investigating ice-atmosphere interactions. J. Geophys. Res., 98, 10085-10109.

Hunke, E. C., and W. H. Lipscomb, 2010: CICE: the Los Alamos sea ice model documentation and software user's manual, version 4.1. LA-CC-06-012.

**6 II. 141-142 I appreciate that you specifically mention the novel aspects of your study. However, I find these aspects only partly novel. The first aspect is not truly novel. Olonscheck and Notz, 2017 (Consistently Estimating Internal Climate Variability from Climate Model Simulations. Journal of Climate) distinguish changes in the variability of winter and summer Arctic sea ice area. The second aspect is also touched by Olonscheck and Notz, 2017 but your study goes beyond this by investigating the underlying processes for the model-simulated changes in CESM-LE. However, the very recent study by Massonnet et al., 2018 (Arctic sea-ice change tied to its mean state through thermodynamic processes. Nature Climate Change) covers parts of your findings. I recommend to more clearly work out the novel aspect of your study, to distinguish your results from the mentioned studies, and to discuss your results in the context of their findings.**

Thank you for pointing us to these very recent papers, which are relevant for and complementary to our study. The Massonet et al. (2018) analysis only addressed sea ice volume variability and its projected future change, which is to be a monotonic decrease, unlike the more complicated time-dependent response of ice area variability that is our focus. Our findings help resolve the findings in Olonscheck and Notz (2017), whose analysis is of future changes in sea ice variability is much coarser temporally and seasonally (the 2006-2100 time block for JAS and JFM), and less conclusive: (a) "the future internal variability of summer Arctic sea ice area possibly increases...", and (b) "... winter and summer Arctic sea ice area show inconsistent model responses" [in CMIP5]. They also identify a decrease in future sea ice volume variability, like Massonet et al., but that is not directly pertinent to our findings.

We have added these two new studies to the introductory material, where we highlight previous findings on this topic and distinguished our focus from their work. In addition, we now compare and contrast our results with those of Olonscheck and Notz in the final section. [Lines 99-104, 408-412]

**7 II. 147-148 I don't believe that the internal variability is robustly characterized from just one model. The internal variability largely differs between the CMIP5 models. How do we know that CESM-LE is representative? I assume you mean that 40 ensemble members allow to robustly quantify the internal variability WITHIN THAT MODEL, but I don't believe that your statement is correct as it is now. Please be more precise here.**

We agree that the sentence needed refining, so we have modified it to address the reviewer's point. The sentence now reads, "We analyze a large 40-member ensemble from a single GCM, which allows us to isolate internal variability, which is otherwise muddled with inter-model variability in multi-model comparisons." [Lines 113-115]

**8 II. 162 A medium ensemble of 15 members for RCP4.5 described in Sanderson et al., 2015 (A new ensemble of GCM simulations to assess avoided impacts in a climate mitigation scenario. Climatic Change) and recent ensembles, e.g., for RCP2.6 described in Sanderson et al., 2017 (Community climate simulations to assess avoided impacts in 1.5 and 2C futures. Earth System Dynamics) also exist.**

**We have added the two Sanderson et al. papers, as suggested. [Lines 132-133]**

**9 II. 170-172 For two reasons, I am not convinced by the usefulness of this selection criteria. First, because the threshold of 20-percent error seems arbitrary to me. How is this justified? Second and more importantly, there is no reason to believe that models that fit the observations comparatively well are better than others because of the large influence of internal variability. When taking model-specific internal variability into account the sea-ice simulations of most CMIP5 models are plausible. I would like to see whether or not your basic conclusions change when using the full set of CMIP5 models. Also, as a reader I would like to know which CMIP5 models you used without having to look this up in Wang and Overland, 2015.**

The model selection criteria was introduced by Wang and Overland 2009 (GRL) and then gradually accepted by the community, not only for sea ice but also for other variables. We want to use trustworthy models, which can capture the corrected physics and dynamics. This is especially important for a sea-ice model, because we are dealing with the absolute value (the sea-ice extent) instead of anomalies (e.g. global mean temperature). The systematic biases in a model can significantly contaminate the ensemble means. Besides, as Massonet et al., (2018) pointed out, models far outside the observed base state should be omitted, because they won't capture the correct physics of thermodynamic processes in the future. How do we define "far outside", and which ones should be omitted? The 20% threshold was chosen because it can remove the outliers yet keep a reasonable number of models for a proper ensemble size. We also tested other selection criteria and determined that 20% is a good choice for this purpose. In IPCC AR5, this model selection method was adapted with a variation for the first time in IPCC history (IPCC, 2013, Chapter12). The 20% cut-off was also followed in their modified selection scheme.

**Model names have been added into the text. [Lines 141-144]**

**10 II. 176-177 To calculate the statistics for each of the 33 ensemble members and to then average them gives a biased estimate, because models with more ensemble members have a larger weight than models with only few (or even one) members. Again, I would like to see whether your basic conclusions would change when you always use e.g. three ensemble members from a model. As it is, I don't find the approach convincing.**

We use no more than 5 ensemble members from each model, even if there are more ensemble members available, to avoid overweighting certain models. In addition, we tried to make the total ensemble number as close as possible to that used in the CESM-LE. This sentence is now added to the text. It is unfortunate that some models only provided one single realization. If we had only kept one member each, then the total ensemble numbers would be too small (12). [Lines 145-147]

**11 II. 212-217 I find the analysis of the CMIP5 models rather weak. To me, it is no**

proof that the variability is indeed increasing as shown by e.g., Goosse et al. 2009. This is because I see no logic behind simply averaging the CMIP5 models. As you write, the timing of ice retreat is very different in the different models, so averaging them will smooth out possible signals. For instance, one could normalize the timing of sea ice retreat before doing the analysis. I think that more analysis of the robustness of the results based on the CMIP5 models is needed.

We appreciate this suggestion and recognize that there is no ideal way to assess the collective response of the CMIP5 models. We are not sure how to normalize the timing of sea ice retreat before doing the analysis. The paper acknowledges that multi-model averaging leads to smoothing, but this procedure is standard in analyses of CMIP5 (e. g., the IPCC AR5 report Chapters 9 to 12). The purpose for presenting the CMIP5 models is to supplement our primary, in-depth analysis of CESM-LE by showing that their first-order features are similar and therefore that our major conclusions are not model-specific. That this resemblance emerges despite the timing differences among the models is evidence that our primary conclusions are robust.

**12 II. 230-232 Related to the previous comment, I would like to know which of the two reasons is more relevant.**

The discussion in the original manuscript was a little confusing, when we say "CMIP5 model spread could also be responsible for inflated variance as models diverge in their timing of the downward trend and its rate of decline". We were trying to explain two different things in one sentence. Here is the revised portion:

"Near the end of the 21st century, the running standard deviation also shows an increase in the CMIP5 ensembles from December to June (Fig. 3), very similar behavior to that displayed by CESM-LE. However the magnitude of the increase in the running standard deviation in the CMIP5 ensemble mean is smaller than that in CESM-LE. This is not surprising, as the timing of ice retreat varies among models, so averaging them will smooth out the possible signals. The CMIP5 models therefore provide additional evidence that increased variability is associated with decreasing sea ice coverage. " [Lines 197-203]

**13 II. 219-220 See again Olonscheck and Notz, 2017.**

We have added comparison of our findings to Olonscheck and Notz in a couple of places, but their analysis of the difference in sea ice area variability averaged between entire time blocks (1850-2005 vs. 2006-2100) doesn't lend itself to a direct comparison with the time series presented in Figures 1 and 2.

**14 II. 304-322 It is not very clear to me how exactly you calculate the thermodynamic and dynamic component. For instance, do you sum up top, basal and lateral melt for the thermodynamic melt component? I think I can guess what you did, but it is not written down precisely.**

The thermodynamic component is a sum of these three terms, as we have clarified in the text. [Lines 265-268]

**15 I. 373 I recommend one or two introductory sentences here to guide the reader. This would also help to improve the structure of the discussion section.**

Good suggestion. We have added some introductory sentences that summarize the overall study and segue to the bullet points in the rest of the section describing our major findings. [Lines 325-329]

**16 I. 443 This should be "projected", instead of "predicted".**

Text has been changed to "projected."

**17 II. 448-449 I very much appreciate that your work includes the analysis of CMIP5 models. But I question that the presented analysis is sufficiently well done to justify this statement on robustness. Especially, because the CMIP5 models are only used for section 3.1 and not in the later sections that deal with the mechanisms. The questionable (see comment #11) and generally weak inverse relationship between variability and rate of retreat that you show for the CMIP5 models does NOT necessarily imply (and also does not suggest) that the same mechanisms are at work like the ones you describe for CESM-LE. This statement is too strong. I recommend to either extend your analysis of the mechanisms to the CMIP5 models (if possible) or further weaken or delete this statement.**

We have softened this sentence by stating that the physical mechanisms identified in CESM "may apply more generally" [to CMIP5 models]. We agree with the reviewer that our conclusions derived from CESM don't demonstrate robustness across all models, but we think that other simulations should exhibit a similar non-linear (parabolic) ice variability response, based on the geographic dependencies described in Goosse et al. (2009) and Eisenmann (2010). [Lines 414-416]

technical corrections:

I. 331 I prefer "the variability in the thermodynamic term", rather than "the thermodynamic term variability"

**The text has been changed as suggested. [Lines 288-292]**

Figures: I suggest to make the figures look more consistently, i.e. Figures 3 and 6 like Figures 1, 2 and 4. Also, I find the different axis labeling in Figure 6a and 6c confusing. For Figure 6, a title for each panel would increase the readability and lines at 0 percent and 100 percent in panels a and c, too.

We have made these modifications to Figure 6 (new Figure 7), though it is unclear otherwise how the figures look inconsistent. The former Fig. 3 and S1 were adjusted to have matching sequential color schemes, while new Fig. 7 is displaying the data in a different way than the other figures and is presented as best as possible. These three figures are included in this discussion comment.

I. 345 frazil = frazil ice?

Yes, frazil "ice" has been added. [Line 306]

References: Comiso et al ... The year of publication is missing. Zhao et al., 2018 ... This reference appears twice.

These two references have been corrected. Thanks for catching these oversights.

**Anonymous Referee #2**

This study by Mioduszewski and co-authors is concerned with the future (and to some degree past and present) variability of the Arctic sea ice cover in GCMs. The article focuses on seasonal aspects of the variability in sea ice area, and on potential drivers of such variability. The authors find a strong correlation between ice area variability and ice thickness, and argue that thermodynamic processes have a stronger impact on variability than dynamic processes.

The study is concerned with an important topic that fits well within the scope of The Cryosphere. I agree by-and-large with the comments of the other reviewer, and would hope to see some substantial revisions of the manuscript. Furthermore, **several parts of the manuscript are marked by a somewhat disappointing standard of language and presentation, in particular given the experience and seniority of the co-authors**.

We have revised the manuscript accordingly, as described in our point-by-point responses below, many of which directly pertain to language and presentation, especially those listed under "Specific comments".

**General comments:**

1. The abstract and introduction should be thoroughly revised (see specific comments below). The writing improves from Section 2 onward.

See our responses below to the reviewer's specific comments, including the abstract and introduction.

2. Please consider the geographic muting effect of Eisenman (2010) in more detail. i.e. what do analogues to Fig S1 and Fig 1 look like when using Eisenman's "equivalent

ice extent"? This would help quantify the role that the distribution of land around the Arctic basin plays in this context.

We have attached this analysis to this discussion comment. The adjustment for geographic constraints does result in some of the expected changes, as there are steeper declines in some of the winter and spring months. However, it shouldn't change the main results shown in the monthly time series of ice variability in Figure 1. Furthermore, while this alternative metric does produce much larger reductions in future ice cover during winter-spring, it's not clear how this result should affect our existing interpretations. Equivalent ice area is a theoretical construct and our purpose is to assess the variability of ice cover that actually exists and that has practical implications for societal impacts such as marine navigation.

We do believe it is worth noting how the calculation of equivalent ice extent compares with our analysis, since this is a relatively well-known concept in the field and some others will likely have the same question, and therefore we have made these modifications to the discussion. [Lines 339-356]

3. I share the concerns of the other reviewer in that the discussion of the CMIP5 analysis is somewhat vague and incomplete. It also should be put more clearly in context with other recent work on the subject.

**Please see our responses to Reviewer 1 about their points #9 and 10, copied below:**

The model selection criteria was introduced by Wang and Overland 2009 (GRL) and then gradually accepted by the community, not only for sea ice but also for other variables. We want to use trustworthy models, which can capture the corrected physics and dynamics. This is especially important for a sea-ice model, because we are dealing with the absolute value (the sea-ice extent) instead of anomalies (e.g. global mean temperature). The systematic biases in a model can significantly contaminate the ensemble means. Besides, as Massonet et al., (2018) pointed out, models far outside the observed base state should be omitted, because they won't capture the correct physics of thermodynamic processes in the future. How do we define "far outside", and which ones should be omitted? The 20% threshold was chosen because it can remove the outliers yet keep a reasonable number of models for a proper ensemble size. We also tested other selection criteria and determined that 20% is a good choice for this purpose. In IPCC AR5, this model selection method was adapted with a variation for the first time in IPCC history (IPCC, 2013, Chapter12). The 20% cut-off was also followed in their modified selection scheme.

We use no more than 5 ensemble members from each model, even if there are more ensemble members available, to avoid overweighting certain models. In addition, we tried to make the total ensemble number as close as possible to that used in the CESM-LE. This sentence is now added to the text. It is unfortunate that some models only provided one single realization. If we had only kept one member each, then the total ensemble numbers would be too small (12). [Lines 145-147]

4. In Sec 3.3 it seems odd to choose Sept and Dec as months to study the roles of ice retreat and expansion, respectively. First, the sea ice minimum occurs typically in mid-Sept, which means that there's substantial ice expansion in the 2nd half of the month (as remarked in L.314). Thus, if the authors want a fully retreating month, why not choose August? December, on the other hand, is fairly early in the ice expansion phase, so if the aim is to capture as much as possible of the preceding expansion, why not choose February? Or January? To that point, in the conclusions (L.413-414) the authors relate Fig 6c,d to "Nov-Jan" variability (rather than to Oct-Dec, as used in the analysis).

We understand why the reviewer questions our choices of representative months, but we tried to balance various considerations in selecting September and December. Because there has been so much interest in September sea ice coverage as the time of annual minimum and previous work on interannual variability in that month (e. g., Goosse et al. 2009, Swart et al. 2015), we felt that a focus on September would be of interest to readers. In addition, our choices for selected months had to strike a balance between conditions in the present climate---in which there is some ice expansion during late September---and future conditions, when September becomes ice-free and therefore no longer has any ice expansion.

December is of particular interest because that is a month of ice expansion exhibiting the distinct three-phase evolution of interannual ice variability described in the text: essentially flat, then a pronounced peak, followed by a decline (see Figure 1). By contrast, other ice-expansion months we could have chosen, such as January-March, do not reach this three-phase evolution, and March has only a modest increase in interannual variability at the tail end of the simulation (Figure 1). A benefit of choosing September and December in the new Figure 7 is that the analysis sheds light on the physical mechanisms responsible for the three-phase evolution in months with very different thermodynamic and dynamic processes operating.

We're sorry about the confusion regarding "November-January" in the original lines 413-414. This range of months was in reference to the slightly lower mean thicknesses coinciding with the peak in interannual ice variability in these months, as shown in the new Figure 5. To clarify this point in the revised text, we have added that the behavior of Nov-Jan is indicated by our findings for December in a new Figure 7c,d as a representative month. [Lines 376-377]

5. As pointed out by the other reviewer, this work needs to be put carefully in context with the very recent paper by Massonnet et al ("Arctic sea-ice change tied to its mean state through thermodynamic processes", Nature Climate Change, 2018). I appreciate that the latter study was published after this one was submitted.

We thank both reviewers for bringing the Massonnet et al. study to our attention, and we point to our response above to comment #6 of Reviewer 1 for how we put our work into that context. While sharing thematic similarities with our paper, Massonnet et al.

focused exclusively on sea ice volume, rather than ice area, which is the focus of our analysis. This distinction is very important, because Massonnet et al. found that the interannual variability of Arctic sea ice volume has already peaked and that it will decline in accordance with the thinning ice pack in the future. By contrast, our study reveals a more complicated behavior in the future interannual variability of sea ice area, such that it exhibits a two-to-three phase evolution in each month: relatively steady during the thick-ice regime of the past, a pronounced increase when the ice packs thins sufficiently, and then a decline if the ice cover diminishes sufficiently in a particular month (July-December). [Lines 98-100]

**Specific comments:**

I would suggest moving Fig S1 to the main text as Fig 1. I'd also suggest color-coding the different months sequentially in this figure so that the seasonal cycle becomes more visible.

This is a good suggestion. We have incorporated this figure into the text as Fig. 1 and changed the color scheme as suggested, which now matches new Fig. 4.

L.49: I would suggest deleting "independent".

We appreciate the suggestion but feel that this term is a useful reminder to readers that the differences among the 40 CESM-LE realizations express purely internal variability within the climate system.

L.52-54: This sentence is somewhat confusing. Some months see an essentially monotonic increase and it's not immediately clear what part of the sentence refers to CESM-LE and what to CMIP5. I suggest rephrasing and/or splitting into 2 sentences.

We realize that this sentence may be ambiguous, so we have reworded into two sentences. The text now states, "Both CESM-LE and CMIP5 models project that ice area variability will indeed grow substantially, but not monotonically in every month. There is also a strong seasonal dependence in the magnitude and timing of future variability increases that is robust among CESM ensemble members." [Lines 36-39]

L.55: "inversely" correlated. This is used at several points in the ms. Unless I'm mistaken, isn't the rate of retreat "directly" correlated with the variability? In other words, the larger the rate of retreat, the larger the variability (?). At L.428 the authors talk about the rate of change. Here I can see the inverse relation: the more negative the rate of change, the larger the variability.

Good point. The reviewer is correct that the variability is directly correlated with the ice retreat rate, so the text has been modified accordingly. We had been thinking in terms of the rate of change in the ice area, which is inversely correlated with the variability.

L.58: "...indicating that [for most of the years (?)] substantial future thinning ..."

We think that a qualifier is unnecessary in this sentence, because we are really just making a general statement that the ice pack needs to thin substantially before a peak in ice variability can be expected.

L.59-60 "... depends on the season, primarily due to whether ..." This could be written more clearly.

That phrasing has been reworded to "... depends on the season, especially whether..." [Lines 43-45]

L.98 "... reduces the [mean] thickness of the basin ice back ..."

Sentence changed as suggested [Lines 58-59]

L.100 " ... the [estimated] negative trend ... "

Sentence changed as suggested [Lines 60-61]

L.103 "[Output from] many climate models suggest[s] ..."

Sentence changed as suggested [Line 63]

**L.113-114 rephrase**

The sentence has been rephrased as follows: "Nonetheless, navigation through the Arctic has already increased in frequency as a result of this decline (Melia 2016; Eguíluz et al. 2016), and even more trade routes associated with the increased ice-free season are expected by the end of the 21st century (Aksenov et al. 2015; Stephenson and Smith 2013)." [Lines 71-75]

L.117 I agree with the other reviewer that the implied causality between reduced extent and loss of multi-year ice is misleading.

As noted above, we have rephrased this sentence: "As the Arctic sea ice pack thins and retreats, multi-year ice is being lost and there is consequently a larger proportion of seasonal, first year ice." [Lines 77-79]

L.118 "Increased thin ice ...". Replace with "Overall thinner ice ..."

Sentence changed as suggested [Line 79]

L.121 "... ice growth and retreat rates ..." I'd argue this should either be "expansion and retreat" or "growth and melt"

The sentence has been changed to, "..., at least partially due to enhanced ice growth and melt." [Lines 81-82]

L.129 "relationship between ice area and its variability". Do the authors mean the "mean ice area" and the "variability in ice area"?

The wording of this topic sentence has been revised for simplicity and to accommodate the broader investigation of sea ice volume variability cited later in the paragraph. This sentence has become, "Changes in the interannual variability of sea ice have been studied only in a limited capacity, likely because they are only beginning to become visible in September in the present day." [Lines 89-90]

L.130 "... it is only beginning to become visible ..." The relationship is becoming visible? Does this mean that we are starting to observe a correlation between the mean ice area and the variability of ice area? Please clarify.

See the comment above.

L.143 "... monthly differences are [societally/economically?] important ..."

We have altered the text to read "societally important" [Line 109]

L.148 "... characterize internal variability [of CESM]" (see other reviewer's comment)

This sentence has been modified as suggested: "We analyze a large 40-member ensemble from a single GCM, which allows us to isolate internal variability, which is otherwise muddled with inter-model variability in multi-model comparisons." [Lines 113-115]

L.198 "... follows [an analogous] three-phase progression ..."

Sentence changed as suggested [Line 169]

L.222 "inverse" see comment at L.55

Sentence changed as described above

Fig.3 I find it hard to decipher the individual curves here. What about splitting the figure into 2 panels, with panel (a) showing spring/summer months and (b) showing fall/winter months. The missing curves in each panel could shown as faint gray in the background for reference. Again, I would use a sequential color map.

We have modified the new Figure 4 by using a sequential color map to help readers discern the individual monthly curves, while still allowing for a direct comparison of the curves on a single graph.

L.240 "... between ice thickness and [ice area] variability ... " Otherwise it might be read as "ice thickness and ice thickness variability"

Sentence changed as suggested [Lines 209-210]

L.267-272 Would it be worth showing another thickness curve (<0.2 m) in Fig 4 to illustrate the phase dependence (and different area coverage) for different ice thicknesses?

We appreciate this suggestion and have tried overlaying another thickness curve on Figure 4, but this ended up cluttering the figure and distracted from the main point we're making with these graphs.

Fig 5: The left hand side of the bounding boxes was cut off. Also, the resolution of the figure was low (jpg? Better to use png with resolution > 150 dpi). It'd be nice to add the respective decade in the top left corner of each panel.

We have resized the figures, so that the left-hand side doesn't get truncated during the online pdf conversion, and we added the decade to each panel.

L.275: Please mention CESM-LE in the caption.

Caption changed as suggested

L.287: "... thin ice and [the variability of] inter-annual ice coverage ... "

**Sentence changed as suggested [Lines 247-248]**

L.280-282: How much of this difference is simply due to the limited run length of the simulations? In other words, once the ice retreats further in winter and spring after 2100, would one then also see the horse-shoe pattern in those seasons? Conversely, in the summer, are the regions of high variability restricted to the Arctic boundaries during earlier decades in the simulations?

Presumably, the horse-shoe pattern of maximum ice variability would shift to the winterspring months after 2100 if the model had been run longer, but we can't prove that with the existing simulations. We have confirmed that the regions of highest variability during summer in the earlier decades of the simulation occur along the periphery of the ice pack, where the thinnest ice exists.

L.293 "expanding" rather than growing

**Sentence changed as suggested [Line 254]**

L.304 Why is a different method used here to calculate the standard deviation?

We have clarified that the method is not actually different, but the decadal average of the running standard deviation is used in new Figure 6, which results in a slightly lower amplitude of standard deviation when comparing to the time series. [Line 265]

**L.337-340 delete "rather than melt". Split into two sentences?**

As suggested, we have deleted that phrase and split into two sentences: "From the 20th century well into the 21st century, ice growth occurs in the October-December period in a similar region of maximum interannual variability as September, except slightly equatorward (Fig. S2b). Ice export plays a relatively larger role in the regions of interest in December than in September (Fig. 6c)." [Lines 298-301]

L.342 "[mid] 21st century"

Sentence has been changed to "early-mid 21st century". [Line 303]

L.345 "frazil [ice]"

Sentence changed as suggested [Line 306]

Fig 6: The (a)-(d) labels are too big and bold, and the rest of the text in the figure is too small.

This figure has been modified to improve readability.

L.376 Isn't the smaller magnitude of spring variability just a result of the time series ending in 2100 (before the ice edge retreats into the Arctic basin in spring)?

Presumably, although we can't prove that. We have modified the sentence by adding "by the time the simulation ends in 2100" to avoid implying that future variability during spring will necessarily be less than in other seasons. [Line 335]

L.415 "... ice area variability [in winter] also coincides ..."

The sentence has been altered nearly as suggested, but we use the phrase "in these months" to refer to the Nov-Jan period cited in the previous sentence (since November isn't a winter month). [Lines 378-380]

---

## Author Response (AR3)

Referee 1:

The revised version of this manuscript is certainly improved. This includes the overall quality of writing and presentation, as well as how the novel aspects of the present findings are highlighted and the discussion of how this study fits in with the existing literature. However, I still have a couple of major concerns (and several more minor ones) that I would like to see addressed. These are detailed below:

Major comments:

1) Even though the authors mention the recent study by Massonnet et al (2018) in the introduction, they basically dismiss it as having no bearing on the present study since Massonnet et al primarily consider sea ice volume, not area. However, volume and area are closely related (through thickness of course), and both Massonnet et al and the present study consider the role of ice thickness in some detail, as well as the main driver of changes in thickness: thermodynamic processes. I therefore believe the present findings should be put in context with those of Massonnet et al. Furthermore, I similarly believe it at least worth a cursory discussion of why Massonnet et al would find sea ice volume variability to decrease monotonically, while sea ice area variability increases initially.

Upon further reflection, we better understand the reviewer's point that the Massonnet et al. study has more relevance for our findings than we originally supposed. Therefore, we have added new material to the Discussion and Conclusions (Section 4) that compares our results with those of Massonnet et al. First, we added the Massonnet et al. reference to the existing Holland et al. (2006) reference for the statement that a higher efficiency of open water formation occurs in conjunction with thinning sea ice [Line 379], which was one of the key findings of the Massonnet et al. analysis. Second, the original final paragraph has been split into two parts to allow elaboration of how and why Massonnet et al.'s findings of decreasing sea ice *thickness* variability with a thinning ice pack contrast with ours that sea ice *area* variability will increase (at least transiently). This new structure offers an interesting contrast and a warning that expectations of future Arctic sea ice variability depend critically on whether ice area or thickness is being considered. [Lines 427-436]:

"Interestingly, another recent study (Massonnet et al. 2018) revealed that CESM-LE simulates a future *decrease* in interannual variability of sea ice *volume*, due to the dominance of the sea ice thickness term. Contrary to the behavior of ice area variability analyzed here, their analysis showed that interannual variability of ice thickness consistently declines when the ice pack thins. This relationship is a robust thermodynamic consequence of a strengthened "ice-formation efficiency", indicative of an enhanced stabilizing ice thickness-ice growth feedback (Notz and Bitz 2015) caused by greater wintertime vertical ice growth following summers with pronounced ice thinning. Therefore, it is important to distinguish which term (area or thickness) is being considered when assessing future changes in the variability of the ice pack."

2) Each time I read the manuscript I struggle to assess how much the differences in the variability evolution between months are just due to the winter months starting at much larger ice cover? I just realized (and apologize that I didn't earlier) that the plot I would really like to see is a scatterplot of standard deviation vs retreat rate, with different symbols for different months (and maybe color coded by time?). How good is the correlation for the individual months (a correlation that is a main finding highlighted in the abstract, L37)? Do all months fall on the same curve?

We thank the reviewer for this helpful suggestion. To elucidate the relationship among months and time periods, we have added two supplementary figures that illustrate the evolution of sea ice variability vs. sea ice loss.  Figure S1 plots the time-evolving relationship by color-coding the smoothed annual data from Figure 2 across successive 30-year time periods in all months from 1920-2100. These graphs indicate that the positive correlation between ice variability and ice loss emerges clearly during the 21st century---even to some degree during the spring months---but the precise relationship is somewhat muddled by the large number of data points.  To help clarify the relationship, Figure S2 (top) displays the average value of the data points from Figure S1 within each 30-year interval for every month.  Again, the emergence of a positive correlation between ice variability and ice loss is apparent in most months, but this relationship is very muted during the spring (and eventually disappears when the ice pack melts off in certain months).  To synthesize all of these individual monthly relationships, Figure S2 (bottom) shows all of the monthly data points on a single graph.  This plot indicates that there is very little relationship between the two variables during the early-middle 20th century (dark blue and aqua points), before the warming signal takes hold, but the correlation is very evident starting in the late 20th century. From that time onward, the strength of the relationship is very stable among time periods (excluding the three points in the late 21st century when the ice pack has disappeared in August-October), as reflected by the similar magnitude of the regression among time periods.

This additional analysis supports our conclusion the correlation between ice variability and ice loss is robust, once sufficient warming occurs to manifest the relationship.  Most months clearly reach this condition sometime during the 21st century, but April and May are exceptions, presumably because the simulation has not run long enough to achieve this state.

The reviewer correctly notes, however, that the statement in the abstract was too sweeping, because it didn't account for the springtime exception.  We have reworded that sentence accordingly, so that it now reads, " The variability generally correlates with the average ice retreat rate, before there is an eventual disappearance in both terms as the ice pack becomes seasonal in summer and autumn by late century." [Line 39]  In addition, we have referred readers to these supplemental figures in Section 3.1 [Line 200].

Minor comments:

L43 I would feel more comfortable if this statement was qualified somewhat. Maybe something like "Our findings suggest that thermodynamic..." or "Our results agree with previous findings that thermodynamic ..."

We have modified the wording as suggested. It now reads, "Our findings suggest that thermodynamic melting . . . " [Line 45]

L72 Please rephrase: this sounds like there won't be more trade routes opening until the end of the century, where in fact the Northeastern Passage was just travelled by a MAERSK container ship a couple of weeks ago.

We have clarified by changing "by" to "throughout" in that sentence. It now reads, ". . . even more trade routes associated with the increased ice-free season are expected throughout the 21$^{st}$ century." [Line 77]

L83 Thin sea ice is not only more vulnerable to atmospheric forcing because of thermodynamics - it is also more easily deformed dynamically (e.g., ridged or opening leads).

Good point. We have incorporated this suggestion into the revised sentence. It now reads, " . . .thin ice is more vulnerable to anomalous atmospheric forcing and oceanic transport due to the smaller amount of energy required to completely melt the ice (Maslanik et al. 1996, Zhao et al. 2018) <and deform the ice dynamically (Hibler 1979)>." [Line 90]

L87 "... sea ice EXTENT have been ..."

We thank the reviewer for catching this typo. The sentence now reads, " Changes in the interannual variability of sea ice <coverage> have been studied only in a limited capacity. . . ". [Line 94]  We prefer "coverage" over "extent", because some studies we cite have used sea ice area and others have used sea ice extent.

L100 " ... much coarser ..." Coarser than who? The present study? Please clarify.

We have clarified this comparison.  The sentence now reads, ". . . [the Olonscheck and Notz (2017)] analysis was much coarser temporally and seasonally <than our study>,  . . . " [Line 108]

L101 " ... between entire blocks of time ..." change to " ... between two discrete time periods ..."

Good suggestion.  The sentence has been changed accordingly. [Line 109]

Fig 4 and associated text, in particular L206-217: I must admit that after reading and re-reading the text multiply times and staring at the figure for a while, I still don't quite understand what is being said/shown. Maybe this can be rewritten more clearly. At this point my understanding is something like "The increase and decrease in variability in September is matched most closely by the increase and decrease in ice that is 30 cm thick". Is that correct?

We apologize for the confusion surrounding this figure, which admittedly is complicated and challenging to explain. What Figure 4 shows is the strength of the relationship between the variability of interannual changes in basin-wide ice area (same as the standard deviation curves in Figure 2) and "the total area of grid cells with mean ice thickness within a given [thickness] range". In other words, the interpretation of Figure 4 for September would be, "The magnitude of year-to-year variations in basin-wide ice area in September is matched most closely by the amount of ice cover around 30 cm thickness (or between 25-40 cm, as stated in the text)." Early in the simulation, the ice pack during September is so thick that there is very little ice this thin, whereas near the end of the simulation there is virtually no ice of any thickness remaining in that month. However, during the interval of maximum September ice variability in the 2020s-2030s, there is a considerable amount of sea ice within this favorable thickness range.

To clarify this point, we have added wording at the end of the existing sentence: " This peak is associated with the thinnest ice of 0.1 m to 0.2 m from October to January <, indicating that the greatest year-to-year variability of basin-wide ice area in these months occurs when there is the greatest coverage of thin sea ice between 0.1 to 0.2 m thickness.>" [Lines 215-216]

Fig 7: I find it makes the figure harder to read that the ice concentration (in %) and concentration trend (in %/day) are plotted on essentially the same axis, even though they represent very different quantities. I personally think it would be easier to interpret 7a,b if the ice concentration had its own axis for the bottom third of each panel. As it is it suggests that there is a relevant link in the magnitude of how the concentration changes and how the concentration trends change.

We have modified Figure 7 as suggested.

Referee 2:

Thanks for the careful response to the reviewer comments on „Past and future interannual variability of Arctic sea ice in coupled climate models". While the authors mainly explain the approaches used in their initial submission, I would have wished to see some additional research to elaborate on the new aspects of the study, as well as a more elaborated comparison of their results to CMIP5. Although I feel that some chances are missed for a high impact study, I find the detailed response sufficiently conclusive to recommend publication in TC.

I have two minor recommendations that the authors may want to consider before publication:

1. Please be consistent with the terminology by using either „bottom melt" or „basal melt" throughout the manuscript.

We have changed "basal" to "bottom" [melt] for consistency. [Line 280]

2. Please cite the mentioned original studies in addition to Perovich et al (2007). [ll.83-84]

We have added the original studies of Grenfell and Maykut (1977), Maykut (1982), Ebert and Curry (1993), and Hunke and Lipscomb (2010). [Lines 87-88]

[revised manuscript text omitted]

Steve Vavrus 11/16/2018 10:27 AM